
# NDCmitiQ v1.0.0: a tool to quantify and analyse GHG mitigation targets

Annika Günther[1], Johannes Gütschow[1], and Mairi Louise Jeffery[2]

[1]Potsdam Institute for Climate Impact Research (PIK), Member of the Leibniz Association, P.O. Box 601203, D-14412 Potsdam, Germany
[2]NewClimate Institute, Schönhauser Allee 10–11, 10119 Berlin, Germany

**Correspondence:** Annika Günther (annika.guenther (at) pik-potsdam.de)

**Abstract.**

Parties to the Paris Agreement (PA, 2015) outline their planned contributions towards achieving the PA temperature goal to "hold [...] the increase in the global average temperature to well below 2°C above pre-industrial levels and pursuing efforts to limit the temperature increase to 1.5°C" (Article 2.1.a, PA) in their Nationally Determined Contributions (NDCs). Most NDCs

include targets to mitigate national greenhouse gas (GHG) emissions, which need quantifications to assess, i.a., whether the current NDCs collectively put us on track to reach the PA temperature goals, or the gap in ambition to do so. We implemented the new open-source tool "NDCmitiQ" to quantify GHG mitigation targets defined in the NDCs for all countries with quantifiable targets on a disaggregated level, and to create corresponding national and global emissions pathways. In light of the five-year update cycle of NDCs and the global stocktake, the quantification of NDCs is an ongoing task for which NDCmitiQ

can be used, as calculations can easily be updated upon submission of new NDCs. In this paper, we describe the methodologies behind NDCmitiQ and quantification challenges we encountered by addressing a wide range of aspects, including: target types and the input data from within NDCs; external time series of national emissions, population, and GDP; uniform approach vs. country specifics; share of national emissions covered by NDCs; how to deal with the Land Use, Land-Use Change and Forestry (LULUCF) component and the conditionality of pledges; establishing pathways from single year targets. For use

in NDCmitiQ, we furthermore construct an emissions data set from the baseline emissions provided in the NDCs. Example use cases show how the tool can help to analyse targets on a national, regional, or global scale, and to quantify uncertainties caused by a lack of clarity in the NDCs. Results confirm that the conditionality of targets and assumptions on economic growth dominate uncertainty in mitigated emissions on a global scale, which are estimated as 49.2–55.7 Gt $CO_2$eq AR4 for 2030 ([10]th / [90]th percentiles, median: 52.4 Gt $CO_2$eq AR4; excl. LULUCF and bunker fuels). We estimate that 77% of global 2017

emissions were emitted from sectors and gases covered by current NDCs (excl. the USA).

Keywords: Open-access; NDCs; GHGs; Emissions; Mitigation; Uncertainties.

## 1 Introduction

In 2018, the Intergovernmental Panel on Climate Change (IPCC) celebrated its [30]th birthday, and in 2020 climate negotiators intended to come together for the [26]th annual Climate Change Conference (COP 26, Conference of the Parties). These numbers





show that efforts to understand and limit climate change are already on the international agenda for several decades. Due to another global crisis – the global Covid-19 pandemic – this year will see no annual COP, as COP 26 is now postponed until November 2021. At the COPs, international policy to limit anthropogenic climate change and avert the climate crisis that we are living in, and for which we and past generations are responsible (Rahmstorf, 2008; IPCC, 1992, 2014; Hegerl et al., 2007; Rocha et al., 2015), is negotiated. An important outcome of this process is the Paris Agreement (PA; UNFCCC, 2015), in

which Parties set out their long-term temperature goal to keep global warming well below 2°C compared to pre-industrial times, while pursuing efforts to limit it to 1.5°C. The importance of limiting global warming to reduce its negative impacts was already pointed out, e.g., in the IPCC First Assessment Report (FAR; IPCC, 1992), and more recently in several IPCC Special Reports (IPCC, 2018a, 2019a, b). The IPCC Special Report on Global Warming of 1.5°C (IPCC, 2018b) notes that "limiting global warming to 1.5°C with no or limited overshoot would require rapid and far-reaching transitions in energy, land, urban

and infrastructure (including transport and buildings), and industrial systems". Global emissions must peak as soon as possible, and drop by an annual 2.7% in the period 2020–2030 to reach the 2°C temperature goal and even by 7.6% to reach the 1.5°C goal (United Nations Environment Programme, 2019).

Nationally Determined Contributions (NDCs) are the backbone of the PA, in which Parties outline their contributions towards achieving the 1.5–2°C temperature goal, with most NDCs including targets to mitigate national greenhouse gas (GHG)

emissions. A quantification of Parties' mitigation pledges is essential to assess their ambition, and to track whether countries are on course to collectively meet the PA temperature goals. Several studies showed that the current set of NDCs are not sufficient to limit global warming even to 2°C (Rogelj et al., 2016; United Nations Environment Programme, 2019; CAT, 2020), and den Elzen et al. (2019) indicated that only six of the G20 members (including China and India) are on track to actually meet their unconditional mitigation targets with current policies.

NDCs are dynamic by nature, with regular updates to "ratchet up" ambition over time (UNFCCC, 2015). Updates are requested at least every five years, starting in 2020, reflecting progress in science and technologies, or improved national circumstances. Synchronised with the NDC updates, a global stocktake will be performed every five years, starting in 2023, to assess if countries are on track to limit global warming in line with the PA global goal (UNFCCC, 2015). Estimates of NDC mitigation targets and global pathways are available, e.g., by the Climate Action Tracker (CAT) or Climate Watch (Climate

Watch, b), and several studies presented quantification results for specific countries. However, the quantification tools and extended descriptions of the underlying methods are seldomly publicly available.

We implemented a new open-source tool "NDCmitiQ" (**NDC miti**gation **Q**uantification tool) to quantify GHG mitigation targets defined in the NDCs for all countries with quantifiable targets on disaggregated level, and to create corresponding national and global emissions pathways. NDCmitiQ can be used for the ongoing task of assessing NDCs, e.g., in the global

stocktake, as it is an open-source tool which can easily be updated upon submission of new NDCs, and be run with emissions data from the NDCs or independent comparison data. The intention of this manuscript is to give an insight into the methodologies behind NDCmitiQ, and to show several examples of analyses that can be performed based on the tool's input and output data. Our aim was to implement an open-source tool with a uniform approach and flexible input to quantify national mitigation





targets – including all countries – and to create national and global un- / conditional emissions pathways consistent with the NDCs.

Several challenges to quantifying NDCs arose during the implementation process and will therefore be described. For example, we want to use a uniform approach as far as possible, but many NDCs need country specific information and assessment to properly understand their targets. Which data is best to use for national emissions / population / GDP if not provided in an NDC (Sect. 2.2)? What if a country does not cover its entire GHG emissions (Sect. 2.3)? How can we deal with emissions from the Land Use, Land-Use Change and Forestry (LULUCF) sector (Sect. 2.4.1)? How should national and global emissions pathways be constructed from single data points? How should a targets' conditionality and range (Sect. 2.4.2) be considered?

This manuscript also includes background information on the different mitigation target types together with their equations and input data needed (Sect. 2.1), and on an emissions data set for 1990–2050 that we constructed from the national baseline emissions provided in the NDCs (Sect. 2.2.3). To complete the emissions data from NDCs and for comparison purposes, the time series currently used in the tool are mainly PRIMAP-hist v2.1 (PRIMAP: Potsdam Real-time Integrated Model for the probabilistic Assessment of emission Paths; Gütschow et al., 2016, 2019; Gütschow, 2019; Nabel et al., 2011), and the new data set of down-scaled Shared Socioeconomic Pathways (SSPs, Gütschow et al., 2020). Other data sets could extend the selection. Finally, we present possible use cases for NDCmitiQ and the underlying data in Sect. 3: (i) an analysis of parts of India's NDC, (ii) an assessment of the differences between the emissions data provided in NDCs and our comparison data (PRIMAP-hist, SSPs), and (iii) an analysis of the impact of different quantification options on national and global emissions pathways.

## 2 NDCmitiQ: methodologies and background information

With this work, we introduce a new Python tool to quantify several types of mitigation targets stated in the currently available (Intended) Nationally Determined Contributions (submissions up to 17[th] April 2020; INDCs turn(ed) into NDCs upon a Party's ratification of the PA and no further distinction is made throughout the manuscript). As NDCmitiQ is implemented in Python and publicly available, the tool can be used by researchers and stakeholders. We chose the programming language Python for its code readability, its large user and developer community, and as it can be run on various operating systems with a free software license.

By describing its methodology and the underlying data, we wish to introduce NDCmitiQ, and point towards challenges in the quantification of NDCs' mitigation targets and the room for interpretation in current targets. All quantifications are based on information that we retrieved from countries' NDCs (available through UNFCCC, b, a). The content of submitted NDCs varies strongly from Party to Party (e.g., Taibi and Konrad, 2018; Rogelj et al., 2017), including various types of contributions and requests, such as mitigation pledges, adaptation targets or financial and technological needs. Most of the submitted documents include targets to mitigate national GHG emissions, which are of major importance to reach the temperature goals set out in the PA and are the focus of our study.

In this section, we introduce the different target types that can be analysed with NDCmitiQ (Sect. 2.1), present the general approach to calculate the GHG mitigation targets (Sect. 2.1.2), and explain which information from NDCs is used as input





to the quantifications (Sect. 2.1.3). Time series from non-NDC sources are used additionally as quantification input to create emissions pathways from point data given in an NDC, or if no data are provided, and for comparison purposes. In Sect. 2.2, we present an overview over the time series of emissions, population, and GDP currently considered in NDCmitiQ. This is

followed by details on how we deal with challenges in the quantification process regarding: how to handle mitigation targets that only cover parts of a country's national emissions (Sect.2.3); how to deal with emissions from LULUCF; how to calculate national emissions pathways, and how to aggregate national pathways over several countries or globally per conditionality and range (Sect. 2.4).

## 2.1   Target types

Several types of GHG mitigation targets can be found in the current set of NDCs. As quantifications differ between the target types, a classification of the targets is needed. All target types we differentiate in NDCmitiQ are given in Table 1. For the target types RBY and REI_RBY, reductions are compared to a historical base year, while for ABU, RBU, and REI_RBU, reductions are compared to Business-as-Usual (BAU). BAU emissions are the emissions a country would have, if – starting from a certain year – no further mitigation actions were taken (inactivity scenario). Countries that indicate the year in which

they plan their emissions to peak are not classified specifically, but the information is considered in the national emissions pathways (Sect. 2.4).

### 2.1.1   Classification of target types: type_main & type_reclass

In principle the classification of target types should be simple. If a country states "we will reduce our GHG emissions by 20% compared to BAU", the target classification is RBU. But what if the country also provides a quantification of its RBU target,

which could then be classified as ABS target? To use both information, we introduce two classifications: "type_main" and "type_reclass", which in the given example are RBU and ABS, respectively.

Possible reclassifications are shown in Figure 1: all targets could be reclassified to ABS targets if enough information is provided in the NDC. NDCs stating actions and policies (type_main: NGT) that additionally provide estimates of the mitigation effects of their planned measures can be reclassified to absolute reductions against BAU (type_reclass: ABU). In some NDCs,

targets are given as different types (e.g., relative reduction compared to BAU but also stated compared to a base year), and it is not always clear which are the "official" targets, leaving room for interpretation. This uncertainty could easily be reduced in future NDCs by clear communication. The double-classification does not only provide important information on which countries include quantification data in their NDCs, but is also used in NDCmitiQ to quantify the targets either primarily based on emissions data from the NDCs (use type_reclass), or external data (use type_main).

On global scale, the reclassification of target types shows a large effect: only seven countries are classified as ABS for type_main, while the reclassification based on available emissions data in the NDCs increases this number to 98 (type_reclass), with many reclassified targets for American, African and South-East Asian countries (Fig. 2). The aggregated emissions from countries with RBY base year targets did not change much over the past years (Fig. 3; number of countries with RBY as type_main: 58, and as type_reclass: 37, including all countries considered by the NDC of the European Union, counted as





**Table 1.** GHG mitigation target types considered in NDCmitiQ, together with their abbreviations and one explanatory example per target type.

| Target type | Long name | Example |
|:---:|:---:|:---|
| **ABS** | **ABS**olute target emissions | The mitigated emissions in the target year are aimed to be 500 Mt $CO_2$eq (net). |
| **RBY** | **R**elative reduction compared to **B**ase **Y**ear | The mitigated emissions in the target year are aimed to be 20% lower than our 2010 emissions. |
| **ABU** | **A**bsolute reduction compared to **B**usiness-as-**U**sual | The mitigated emissions in the target year are aimed to be 350 Mt $CO_2$eq lower than our Business-as-Usual emissions in the target year. |
| **RBU** | **R**elative reduction compared to **B**usiness-as-**U**sual | The mitigated emissions in the target year are aimed to be 20% lower than our Business-as-Usual emissions in the target year. |
| **AEI** | **A**bsolute **E**missions **I**ntensity target | The mitigated per-capita emissions intensity in the target year is aimed to be 2.1 t $CO_2$eq/cap. |
| **REI** | **R**elative reduction in **E**missions **I**ntensity compared to a base year OR target year | The mitigated per-capita emissions in the target year are aimed to be 20% lower than our 2010 per-capita emissions (comment: REI_RBY, compared to a base year). OR The mitigated emissions per unit of GDP in the target year are aimed to be 20% lower than our Business-as-Usual emissions per unit of GDP in the target year (comment: REI_RBU, compared to BAU – this option is similar to an RBU target). |
| **NGT** | **N**on-**G**HG **T**arget | We aim on increasing our energy efficiency by 40% (comment: nothing is calculated, baseline emissions are assumed). |

single countries). In recent years, the major share of global emissions with a clear emissions increase, was caused by countries with REI emissions intensity targets, mainly due to the fact that India and China chose REI targets. For NDCs with REI targets, the reclassification of target types does not noticeably impact the global emissions share, pointing towards missing numerical data in the NDCs. The United States of America submitted a formal notification on its withdrawal from the PA to the United Nations on the 4[th] November 2019 (Pompeo, 2019), which took effect on the 4[th] November 2020. Therefore, emissions

from the USA are counted towards "No NDC" and the mitigation measures presented in their NDC are not considered in quantifications throughout this manuscript.

### 2.1.2 Calculating GHG mitigation targets: general equations

In NDCmitiQ we use several equations to quantify GHG mitigation targets, differentiated based on the target types. The equations presented in this section provide important information on the data needed for the quantifications, and allow a first

guess on possible uncertainties connected to each target type. Our general assumption is that the target emissions are the sum of the emissions in the reference year that are subject to mitigation measures (covered), plus the BAU emissions in the target





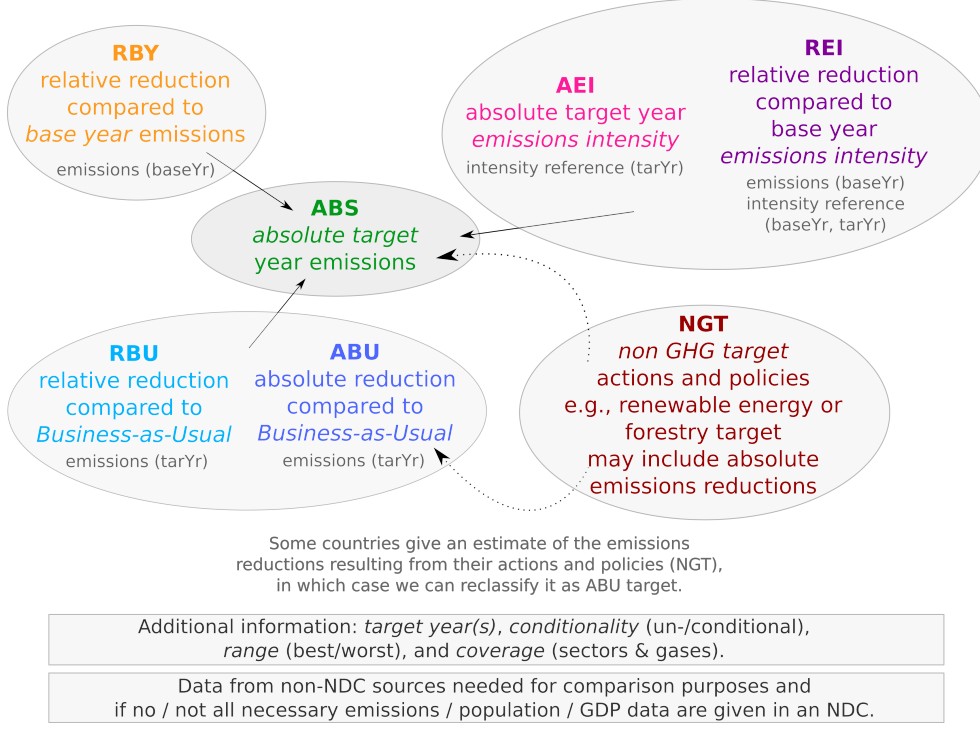

**Figure 1.** Scheme of GHG mitigation target types and possible reclassifications. All targets can be reclassified as ABS, if enough numerical information is provided in an NDC. Additionally, information on the numerical data needed for a target quantification is given.

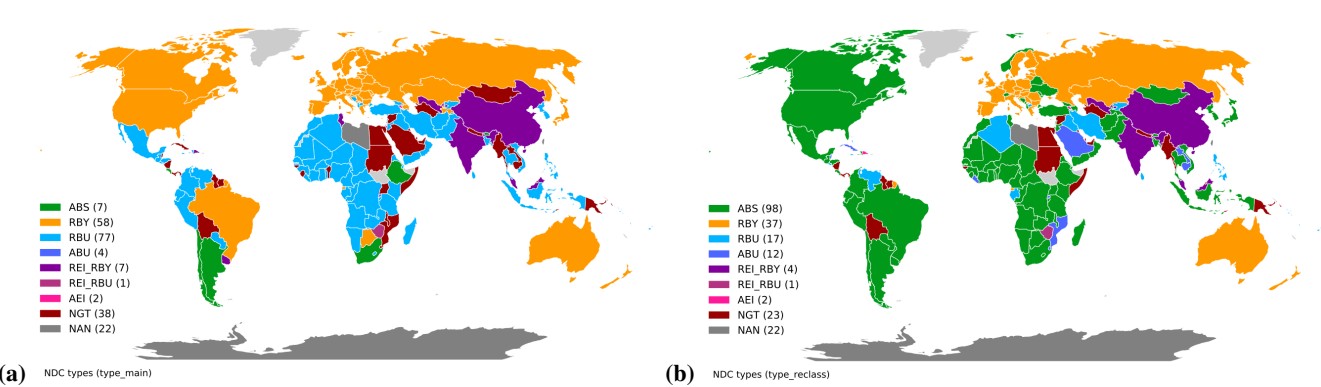

**Figure 2.** World maps presenting mitigation target types as stated in NDCs (panel a: type_main) and reclassified target types – depending on the available numerical information in the NDCs (panel b: type_reclass). In brackets: number of countries with a certain target type (countries that are part of the EU NDC are counted as single countries). USA: not part of quantifications.





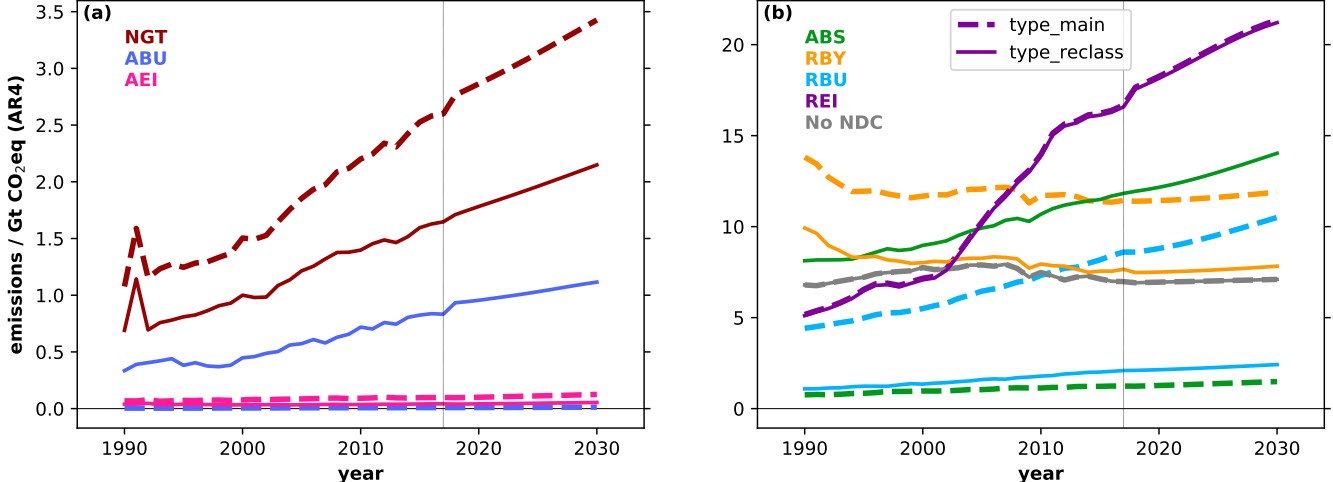

**Figure 3.** Aggregated time series of national baseline (non-mitigated) emissions for all countries with a certain NDC mitigation target type (for type_main & type_reclass). Emissions based on PRIMAP-hist v2.1 and down-scaled SSP2 marker scenario (data source: Gütschow et al., 2019; Gütschow et al., 2020, see Sect. 2.2). Emissions from LULUCF and bunkers fuels are excluded (bunkers fuels: international shipping and aviation; not attributed to individual countries under the United Nations Framework Convention on Climate Change (UNFCCC); Gütschow et al., 2020). A vertical line indicates the last year of historical data (2017). USA: classified as "No NDC".

year from sectors and gases that are not covered and are therefore expected to follow a Business-as-Usual pathway. Unless more detailed information is provided in an NDC, we assume similar efforts across all covered sectors and gases.

In the following, we introduce equations to calculate the target emissions for the different target types assessed in our module, starting with the very similar equations for RBY, REI, and RBU targets. The handling of emissions from LULUCF is not addressed here but in Sect. 2.4. We start with the equation for a relative reduction compared to base year emissions (RBY, Eq. 1).

$$\text{emiTarget}_{\text{RBY}} = \text{NDC}_{\%\text{level}} \cdot \text{emiBL}_{\text{COV}_{\text{refYr}}} + \text{emiBL}_{\text{notCOV}_{\text{tarYr}}} \tag{1}$$

The equation consists of the following elements:

- $\text{emiTarget}$ are the "target emissions".

- $\text{NDC}_{\%\text{level}} = \frac{100\% - \text{NDC}_{\%\text{reduction}}}{100\%}$, with $\text{NDC}_{\%\text{reduction}} = \text{percentage\_reduction\_given\_in\_the\_NDC}$ (e.g., 20% reduction compared to BAU: $\text{NDC}_{\%\text{level}} = \frac{100\% - 20\%}{100\%} = 80\%$).

- $\text{refYr}$ and $\text{tarYr}$ are the "reference year" and the "target year". For an RBY target, the reference year is a historical base year.

- $\text{emiBL}$ are the national baseline emissions, with $\text{emiBL}_{\text{COV}}$ being the share of national baseline emissions covered by the target (depending on the covered sectors and gases), while $\text{emiBL}_{\text{notCOV}}$ is the not-covered share of emissions. $\text{emiBL}_{\text{COV}} = \text{emiBL} \cdot \frac{\sum \text{emi\_from\_covered\_sector\&gas\_combis}}{\text{national\_emissions}}$. The percentage reduction (here as $\text{NDC}_{\%\text{level}}$) is only ap-





plied to the covered share of emissions. $\mathrm{emiBL_{notCOV}}$ stays "untouched" by the reductions and $\mathrm{emiBL_{notCOV_{tarYr}}}$ is therefore added as is.

While for RBY the reference year is a historical year, for an RBU target (relative reduction compared to BAU) the reference year equals the target year, leading to Eq. 2.

$$\mathrm{emiTarget_{RBU}} = \mathrm{NDC_{\%level}} \cdot \mathrm{emiBL_{COV_{tarYr}}} + \mathrm{emiBL_{notCOV_{tarYr}}} \qquad (2)$$

The equation for an REI_RBY target – a relative reduction in emissions intensity compared to the emissions intensity in a historical base year – is also very similar to the RBY target. However, instead of the absolute emissions, the emissions intensity

per-capita or per unit of GDP is reduced. A socio-economic growth factor has to be considered, and $\frac{\mathrm{IntensityRef_{tarYr}}}{\mathrm{IntensityRef_{refYr}}}$ is added (Eq. 3; $\mathrm{IntensityRef_{refYr/tarYr}}$: national baseline population or GDP).

$$\mathrm{emiTarget_{REI\_RBY}} = \frac{\mathrm{IntensityRef_{tarYr}}}{\mathrm{IntensityRef_{refYr}}} \cdot \mathrm{NDC_{\%level}} \cdot \mathrm{emiBL_{COV_{refYr}}} + \mathrm{emiBL_{notCOV_{tarYr}}} \qquad (3)$$

The equations for the remaining target types (ABS, ABU, AEI, and NGT) are given in Eq. 4 to 7. $\mathrm{NDC_{absoluteEmissions}}$ are the target emissions, $\mathrm{NDC_{absoluteReduction}}$ is the absolute reduction, and $\mathrm{NDC_{emissionsIntensity}}$ is the targeted emissions intensity

per capita or unit of GDP, given in the NDC. The given absolute target emissions (ABS) and the absolute target emissions intensity (AEI) are assumed to cover the entire national emissions, else the BAU emissions of the not-covered sectors and gases in the target year would need to be added.

$$\mathrm{emiTarget_{ABS}} = \mathrm{NDC_{absoluteEmissions}} \qquad (4)$$

$$\mathrm{emiTarget_{ABU}} = \mathrm{emiBL_{tarYr}} - \mathrm{NDC_{absoluteReduction}} \qquad (5)$$

$$\mathrm{emiTarget_{AEI}} = \mathrm{NDC_{emissionsIntensity}} \cdot \mathrm{IntensityRef_{tarYr}} \qquad (6)$$

$$\mathrm{emiTarget_{NGT}} = \mathrm{emiBL_{tarYr}} \qquad (7)$$

### 2.1.3 Quantification input per target type and country

Based on the current set of NDCs, we give further insight into the data needed for the quantifications by target type and country. The various target types require different input, as presented in Table 2. Some of the required information must be provided in the NDC: base year; target year; relative or absolute reduction for RBY, RBU, REI, and ABU; absolute target emissions for ABS; absolute emissions intensity for AEI. For a clearly formulated target, the information on which gases and sectors are covered and the share of covered emissions (%cov) in the base and target year should additionally be given in the NDC. Else,

the covered share of emissions relies on assumptions or own estimates. The assumed baseline emissions and intensity reference in the base / target year should be stated in the NDC, but can also be used from "external" sources. A large number of input data requirements does not necessarily imply higher uncertainty, as can be seen for RBY targets: most of it is "easy-to-get", and historical estimates generally have lower uncertainties compared to projections (e.g., BAU emissions). Nevertheless, even RBY targets can be complicated, if, e.g., not targeting all emissions, if no emissions were recorded for the base year, or if the





handling of LULUCF is not clear. Based on our assessment of current NDCs, Figure 4 contains an overview of the emissions, population, and GDP data needed to quantify the targets on country-level, together with the specific years, and target types (type_main). One can also conclude the chosen base and target years from this overview, with the year 2030 being the most prominent target year.

**Table 2.** Input needed for the quantification of NDC GHG mitigation targets, per target type. Some information or data can be retrieved from NDCs only ("NDC"), while for some data "external" sources can be used (indicated in column "Source"). "Coverage" can be the covered sectors and gases or numerical values for the share of national emissions affected by the mitigation target (for the base and target year). "(x)" indicates that the information is only needed for Parties that do not cover all of their national base year emissions.

| What | Source | NGT<br>Non-GHG target | ABS<br>Absolute emissions | ABU<br>Abs. red. vs. BAU | AEI<br>Abs. emi. intensity | RBY<br>Rel. red. vs. base year | REI_RBY<br>Rel. red. vs. emi. intensity | RBU<br>Rel. red. vs. BAU | REI_RBU<br>Rel. red. vs. emi. intensity |
|---|---|---|---|---|---|---|---|---|---|
| Base year | NDC | | | | | x | x | | |
| Target year | NDC | | x | x | x | x | x | x | x |
| Coverage | NDC | | | | | x | x | x | x |
| Relative reduction | NDC | | | | | x | x | x | x |
| Absolute reduction | NDC | | | x | | | | | |
| Target emissions | NDC | | x | | | | | | |
| Target emissions intensity | NDC | | | | x | | | | |
| Refer. emissions in base year | NDC / external | | | | | x | x | | |
| Refer. emissions in target year | NDC / external | | | x | | (x) | (x) | x | x |
| Reference intensity in base year | NDC / external | | | | | | x | | |
| Reference intensity in target year | NDC / external | | | | | x | | x | |

## 2.2 Emissions, population, and GDP data

If an NDC provides enough numerical information, the target quantifications can be based solely on the NDC-data. However, if data are missing, for comparison purposes, and for the construction of emissions pathways, "external" data are needed. Even though the required input varies substantially between Parties (Sect. 2.1.3), in NDCmitiQ we aim at a quantification in an automatic manner. Therefore, the input are time series of national emissions, population and GDP, spanning the period of 1990–2050, and pre-processed in a similar way for all countries.

First, we present the external data currently available in NDCmitiQ (Sect. 2.2.1 and 2.2.2), and then introduce a data set of emissions time series constructed from the baseline emissions given in NDCs (Sect. 2.2.3). This data set can then be used for target quantifications and to derive mitigated emissions pathways. The presented emissions data generally follow the IPCC 2006



**Figure 4.** Reference emissions, population, and GDP data needed for the assessment of current NDCs (type_main). Per country with an NDC, markers indicate that for a certain year emissions (squares), population (circles), or GDP (triangles) data are needed. Crosses indicate target years for ABS and AEI, for which no further emissions data are needed for the quantification. Black boxes indicate data needed if only parts of the national emissions are covered. Only years for which information is needed for type_main are displayed (colour coded; NGTs: shaded red area). EU target: shown for single countries (e.g., Germany). USA: for information purposes only. Vertical dashed lines separate countries with different initial letters.





sectoral categorisation (IPCC, 2006), with few additional categories following the PRIMAP-hist v2.1 nomenclature (Gütschow et al., 2016, 2019). The current implementation of NDCmitiQ is based on the Global Warming Potentials (GWPs) of the IPCC
Fourth Assessment Report (AR4; IPCC, 2007).

### 2.2.1 Emissions data from non-LULUCF sectors, and time series of population and GDP

**Historical data (1990–2017)**

For the quantifications, we need time series of national emissions, population and GDP on country-level, spanning the period 1990–2050. Historical emissions are especially important for targets referring to base year emissions (RBY, REI_RBY). For
the years 1850–2017, emissions data are available from PRIMAP-hist v2.1 (Gütschow et al., 2016, 2019), for different sectors (excluding LULUCF) and gases. The PRIMAP-hist composite data set covers all UNFCCC Parties and most of the non-UNFCCC territories, with complete time series (for more information see Gütschow et al., 2019). We use the data set version in which country-reported data are prioritised (HISTCR: Historical Data Country Reported).

For the quantification of targets with a 100% coverage, emissions time series of national totals are sufficient. However, we
also consider the covered – and not-covered – share of emissions (%cov) and test the influence on the quantification results. To derive estimates of %cov, we use various time series from the PRIMAP-hist data set (for 1990–2017), which differ regarding the contributing sectors and type of emitted gas. While the main quantifications are based on national total Kyoto GHG emissions excluding LULUCF (exclLU; contributions from LULUCF treated separately), more refined time series are used to estimate the covered share of emissions. Therefore, national emissions from the main sectors "Energy", "Industrial Processes and Product
Use" (IPPU), "Agriculture", "Waste", and "Other" are also used (adding up to the national totals exclLU), together with the information on the respective contributions from carbon dioxide ($CO_2$), methane ($CH_4$), nitrous oxide ($N_2O$), and for IPPU the basket of hydrofluorocarbons (HFCs) and perfluorocarbons (PFCs), as well as sulfur hexafluoride ($SF_6$), and nitrogen trifluoride ($NF_3$). The Kyoto GHG basket consists of all the before-mentioned gases.

As for the above described non-LULUCF emissions, the current data source in NDCmitiQ for time series of population
and GDP PPP for 1990–2017 is PRIMAP-hist v2.1 (Gütschow, 2019). Historical population or GDP data are important to derive the socioeconomic growth factor for REI_RBY targets. PPP stands for the Purchasing Power Parity the national GDP is adjusted by for better comparability on international levels (throughout the manuscript we will use "GDP" for GDP PPP). Time series are complete and data are available for all UNFCCC Parties and several additional countries.

**Scenarios (period after 2017)**
For the period after 2017, we use emissions (exclLU), population, and GDP data published recently by Gütschow et al. (2020). In their study, Shared Socioeconomic Pathways (SSPs, available until 2100; Riahi et al., 2017; Crespo Cuaresma, 2017; Dellink et al., 2017; Leimbach et al., 2017) were down-scaled to country-level. The SSP pathways "describe plausible major global developments that together would lead in the future to different challenges for mitigation and adaptation to climate change" (Riahi et al., 2017), and are based on five narratives (Table 3). We chose to include the five marker scenarios in NDCmitiQ,
which were derived using different Integrated Assessment Models (IAMs).





**Table 3.** SSP narratives, mitigation and adaptation challenges, and IAMs for the marker scenarios.

| SSP | Narrative | Challenges for | | IAM for marker scenario |
|-----|-----------|----------------|-----------|-------------------------|
| | | mitigation | adaptation | |
| SSP1 | Sustainability: Taking the Green Road | Low | Low | IMAGE (by PBL) |
| SSP2 | Middle of the Road | Medium | Medium | MESSAGE-GLOBIOM (IIASA) |
| SSP3 | Regional Rivalry: A Rocky Road | High | High | AIM/CGE (NIES) |
| SSP4 | Inequality: A Road Divided | Low | High | GCAM (PNNL) |
| SSP5 | Fossil-fuelled Development: Taking the Highway | High | Low | REMIND-MAgPIE (PIK) |

While for the quantification of national targets, per-country data are needed, the SSP emissions pathways are generally only available for several world-regions. The pathways were down-scaled to the national level by Gütschow et al. (2020), using results from the "SSP GDP [...] country model results as drivers for the downscaling process" (Gütschow et al., 2020). From (Gütschow et al., 2020), we use the data with the source names PMSSPBIE and PMSSPBIEMISC, and the scenarios named SSP1BLIMAGE, SSP2BLMESGB, SSP3BLAIMCGE, SSP4BLGCAM4, and SSP5BLREMMP (BL: baseline). These are SSP IAM scenarios (emissions and socio-economic data), down-scaled using "convergence downscaling with exponential convergence of emissions intensities and convergence before transition to negative emissions", with bunkers emissions having been removed before down-scaling, and data being harmonised and combined with PRIMAP-hist v2.1 time series. The emissions data are national values, excluding LULUCF and international bunkers fuels, available for the gas baskets Kyoto GHG and F-gases (fluorinated greenhouse gases: consisting of HFCs, PFCs, $SF_6$ and $NF_3$), and for the individual gases $CO_2$, $CH_4$, and $N_2O$. In terms of sectoral resolution, only national totals are available. As explained in Sect. A1, pre-processing of the down-scaled SSPs is performed to fill some missing time series for countries with low emissions, population, or GDP. Additionally, for the estimation of %cov and as not all NDCs cover all F-gases, the time series of F-gases are split into the contributing component gases by assuming recent ratios of HFCs, PFCs, $SF_6$ and $NF_3$ (see Sect. A1). The *d*ownscaled time series of the *m*arker scenarios for *SSP1–5* are generally abbreviated as *dmSSP1–5* throughout the manuscript.

### 2.2.2 Emissions data from LULUCF

In the previous section, only emissions data that exclude contributions from LULUCF were discussed. However, for the quantification of mitigation targets, LULUCF emissions are often needed as well. LULUCF is "A greenhouse gas inventory sector that covers emissions and removals of greenhouse gas resulting from direct human-induced land use, land-use change and forestry activities" (UNFCCC, c). As it can be difficult to distinguish the anthropogenic and the natural part of the land-related fluxes, estimating LULUCF emissions is more complex than for non-LULUCF sectors (Joint Research Centre). It is complicated to estimate mitigation effects by LULUCF activities, as gas fluxes depend, i.a., on the age (distribution) of trees. This distribution varies over time – a difficulty not connected to non-LULUCF emissions (Joint Research Centre). LULUCF can further work as an emissions source or sink, can have high inter-annual variability (Fyson and Jeffery, 2019), and data have a





high uncertainty (Roman-Cuesta et al., 2016). As a consequence of the stated problems, we distinguish between LULUCF and non-LULUCF emissions.

For LULUCF, emissions data availability is limited, with some data sources only providing few data points, and as high inter-annual fluctuations are possible in the LULUCF emissions, reasonable gap-filling is difficult. PRIMAP-hist v2.1 does not contain emissions from LULUCF, "due to data availability and methodological issues" (data description document for

Gütschow et al., 2019). Data scarcity and fluctuations also make it complicated to combine data sets, and estimates vary strongly between data sources (PIK).

To choose external national LULUCF emissions data for the target quantifications, several data sets of LULUCF emissions are analysed for available data in the following prioritised order: CRF 2019, CRF 2018, BUR 3, BUR 2, BUR 1, UNFCCC 2019 and FAO 2019 (abbreviations: Sect. A2). For a country, CRF 2019 data are used if available, else CRF 2018, and so on.

Estimates of LULUCF emissions provided by Parties are chosen when possible (similar to e.g., PRIMAP-hist v2.1 HISTCR for non-LULUCF emissions, Gütschow et al., 2019; or Fyson and Jeffery, 2019, for LULUCF emissions). As in Fyson and Jeffery (2019), we include FAO 2019 data, which are calculated using the IPCC methodologies and are based on country-reported data. However, their definitions and data coverage differ (Tubiello et al., 2015) and they "are not directly comparable with UNFCCC data" (Fyson and Jeffery, 2019). As we intend to work with complete time series in order to have complete

emissions pathways up to 2030 or 2050, gap-filling and extrapolation are applied (Sect. A2), neglecting to some extent the challenges with LULUCF emissions described above. Our projections are generally based on constant extrapolation of the average 2010–2017 emissions (more details in Sect. A2).

Globally, in 2017 LULUCF was an estimated net sink of -2.1 Gt $CO_2$eq (GWP: AR4; Table 4), with the estimate based on data that we chose from different sources on country-level (prioritisation as above and details in Sect. A2). In comparison, in

all non-LULUCF sectors a total of 47.6 Gt $CO_2$eq were emitted in 2017 (based on PRIMAP-hist v2.1 HISTCR; excl. LULUCF and bunkers fuels). The aggregated 2030 estimates of net LULUCF emissions (-2.2 Gt $CO_2$eq) are in line with the estimate by Fyson and Jeffery (2019) (-2.0 Gt $CO_2$eq/year). If FAO 2019 is chosen as primary data source, the global 2017 aggregate is an emissions source of +3.4 Gt $CO_2$eq instead of a global sink. As pointed out in Fyson and Jeffery (2019), the choice of the LULUCF data source has considerable effects on the best estimate for the year 2030 in some cases, and the higher

global aggregate of +3.4 Gt $CO_2$eq (2017) when prioritising FAO data is in line with their estimate based only on FAO data (3.3 Gt $CO_2$eq/year, 2004–2014; further comparisons in their study).

### 2.2.3  Emissions data from the NDCs

While the before mentioned data are time series from non-NDC sources, to quantify the targets our intention is to also use the emissions data provided in the NDCs when available. With NDCmitiQ, we are aiming to create national emissions pathways

and global aggregates from the quantified targets. However, in NDCs, emissions are generally given as point data, not counting in data visualisations from which it is often difficult to read the numbers. The external data sources serve to complement the NDC data to emissions pathways, and for comparison purposes. As output from NDCmitiQ, we intend to create mitigated emissions pathways that exclude LULUCF emissions and we therefore construct a data set of national baseline emissions time





**Table 4.** Global LULUCF emissions for several years, resulting from different prioritisations of data sources (default: first row). "Total" are the global emissions, "Net sources" is the sum over the emissions from countries with a net LULUCF emissions source, and "Net sinks" is the aggregate over countries for which LULUCF is a net sink.

| Order of prioritised sources | Total | | | | Net sources | | | | Net sinks | | | |
|---|---|---|---|---|---|---|---|---|---|---|---|---|
| Gt CO$_2$eq AR4 | 1990 | 2010 | 2017 | 2030 | 1990 | 2010 | 2017 | 2030 | 1990 | 2010 | 2017 | 2030 |
| CRF, BUR, UNFCCC, FAO | 0.2 | -2.5 | -2.1 | -2.2 | 3.7 | 3.5 | 3.8 | 3.8 | -3.5 | -6.0 | -5.9 | -6.0 |
| UNFCCC, CRF, BUR, FAO | 0.4 | -2.0 | -1.9 | -2.0 | 3.8 | 3.8 | 3.8 | 3.8 | -3.4 | -5.8 | -5.7 | -5.8 |
| FAO, CRF, BUR, UNFCCC | 4.1 | 3.3 | 3.4 | 3.4 | 5.5 | 5.1 | 5.0 | 5.0 | -1.4 | -1.8 | -1.6 | -1.6 |

series excluding LULUCF (1990–2050) that is based on the NDCs' baseline emissions, combined with PRIMAP-hist and SSP
data for completeness (see Sect. A3).

If available the following baseline emissions data were retrieved from the NDCs: excluding LULUCF ("exclLU"), including LULUCF ("inclLU"), and for LULUCF only ("onlyLU"). As it is not always clearly stated what the provided emissions stand for, some of the classifications are based on a best-guess approach. The emissions estimates are used as long as one can assume that all – or most – of a country's emissions are included.

**2.2.4    Global Warming Potentials**

Emissions throughout this manuscript, and in most of the NDCs, are given as CO$_2$ equivalents, to make emissions from different gases comparable, and provide basket emissions. Emissions in CO$_2$eq follow a certain Global Warming Potential (GWP), with all emissions in NDCmitiQ currently being based on GWPs from the IPCC Fourth Assessment Report (AR4; IPCC, 2006). Inconsistencies can arise when using NDC emissions data (baseline emissions, and ABS, ABU, and AEI targets), which are
partly based on GWPs from the IPCC Second, or Fifth Assessment Report (SAR, AR5; IPCC, 1996, 2014), or unspecified.

To reduce the uncertainty resulting from emissions based on different GWPs, we apply national conversion factors to the NDC emissions data given in GWPs from SAR. The conversion factors are derived from PRIMAP-hist v2.1 HISTCR Kyoto GHG national totals (excl. LULUCF; national averages for the period 2010–2017). Conversion factors are only calculated from SAR to AR4, as from PRIMAP-hist Kyoto GHG emissions are not available for GWPs from AR5, due to missing AR5 data
for HFCs and PFCs.

Global Kyoto GHG emissions in 2017 were 46.3 Gt CO$_2$eq following GWPs from SAR (excl. LULUCF and bunkers fuels), and 47.7 Gt CO$_2$eq for AR4, equivalent to a 2.8% increase in their estimated forcing over a period of 100 years. The higher the national share of CO$_2$ emissions, the lower is the effect of a change in GWPs, as the GWP of CO$_2$ is 1 by definition. A clear communication by Parties of the applied GWPs can reduce this uncertainty in emissions data retrieved from NDCs,
and ultimately in the quantification of the target. We assess 50 / 35 / 5 countries to follow GWPs from SAR / AR4 / AR5, representing 6.9% / 33.7% / 4.2% of global Kyoto GHG emissions (year 2017, excl. LULUCF and bunkers fuels). For the remaining countries we could not retrieve information on chosen GWPs from their NDCs and assume the given emissions to follow the GWPs from AR4.





In the Katowice climate package (Annex to decision 18/CMA.1: Modalities, procedures and guidelines for the transparency
framework for action and support referred to in Article 13 of the Paris Agreement; UNFCCC, 2019), it was decided for the
"National inventory report of anthropogenic emissions by sources and removals by sinks of greenhouse gases" (II) that "Each
Party shall use the 100-year time-horizon global warming potential (GWP) values from the IPCC Fifth Assessment Report, or
100-year time-horizon GWP values from a subsequent IPCC assessment report as agreed upon by the CMA, to report aggregate
emissions and removals of GHGs, expressed in $CO_2$eq" (II.D.37). Implementation of these principles would lead to increased
clarity, and also applying these principles to their NDCs would further increase transparency.

## 2.3 Share of emissions covered by NDCs

In the current set of NDCs, not all mitigation targets cover the total of national emissions. To estimate the uncertainty in target
emissions resulting from different assumptions on the share of covered emissions (%cov), and for comparison purposes, two
options are implemented in NDCmitiQ: use %cov = 100%, or estimates of %cov that are based on the stated targeted sectors
and gases, as described in the current section. Additionally, estimates of %cov indicate which countries have room to improve
their coverage in an updated NDC.

With Article 4.4 of the Paris Agreement (UNFCCC, 2015), Parties to the PA agreed on the following: "Developed country
Parties should continue taking the lead by undertaking economy-wide absolute emission reduction targets. Developing coun-
try Parties should continue enhancing their mitigation efforts, and are encouraged to move over time towards economy-wide
emission reduction or limitation targets in the light of different national circumstances." To reduce uncertainties and increase
transparency, the targets' scope should be defined in NDCs in terms of covered sectors and gases, and in numerical values of
%cov in the historical base year (if needed), and the target year. It should be clear which emissions are targeted by mitiga-
tion actions, or stay "untouched" and are intended to develop under a Business-as-Usual pathway, but not all Parties clearly
communicated this information.

We assessed the NDCs for information on the covered sectors and Kyoto GHGs to estimate %cov, focusing on the main
sectors Energy, IPPU, Agriculture, Waste, Other, and LULUCF, and the single gases $CO_2$, $CH_4$, $N_2O$, $SF_6$, $NF_3$, as well as
the gas baskets of HFCs, and PFCs. In all NDCs we could find some information on targeted sectors – not always clearly
stated, however – and not all NDCs include information on the covered gases, leaving room for interpretation (unclear cases
for sectors: 38 NDCs, and for gases: 27 NDCs). The rules to determine %cov for the national emissions excluding LULUCF are
presented in Sect. A3.1 (results: Sect. 3.1). In general, for years up to 2017, %cov is derived from the PRIMAP-hist emissions
data per sector and gas combination, while estimates for later years are either based on a constant extrapolation of recent %cov,
or on the correlation between national total and covered emissions. Regarding LULUCF emissions, the applied rule is simple:
if the sector is assessed to be covered, its total emissions are assumed to be covered (not taking into account the contributions
of the different gases relevant for LULUCF emissions: $CO_2$, $CH_4$, $N_2O$).



## 2.4 Target emissions and emissions pathways

In order to quantify the Parties' targets, we assessed all NDCs regarding their target types, target years, conditionality and range, covered sectors and gases, and provided emissions. National target emissions are calculated for each target year, conditionality, and range. NDCs include either or both unconditional and conditional targets ("conditionality"), where mitigation actions are conditional upon, for example, international financial support or technology transfer. Some Parties decided to give a range rather than an exact target value (e.g., "unconditional reduction of 26–28%") which we treat here as "best" & "worst", meaning more & less ambitious).

Section 2.4.1 contains information on how we deal with targets that include contributions from LULUCF, how we derive target emissions excluding LULUCF in these cases, and why a separation into $\mathrm{emiTarget_{inclLU}}$ and $\mathrm{emiTarget_{exclLU}}$ is useful. To analyse whether the pledges put us on track to limit global warming to 1.5–2°C, regional or global emissions pathways are needed. Therefore, national emissions pathways that are consistent with the NDC targets for the single target years must be constructed and aggregated. The methodology and options for pathway creation implemented in NDCmitiQ are explained in Sect. 2.4.2.

### 2.4.1 Target emissions: including and excluding LULUCF

LULUCF and its contributions towards a mitigation goal complicate target quantifications (e.g., Forsell et al., 2016; Fyson and Jeffery, 2019; Hargita and Rüter, 2015), which is why, for example the Climate Equity Reference (2018) "dropped support for including LULUCF emissions in the assessment of the NDCs – the quality of the data and of the information in the NDCs simply wasn't good enough to do that with confidence". Reasons for the LULUCF component to be an issue are: there are large uncertainties in the LULUCF emissions data; LULUCF emissions can have high inter-annual variability; as LULUCF can be a net sink, countries can use this sector to disguise increased emissions or missing mitigation ambition in the non-LULUCF sectors; and comparability between national mitigation goals is easier when excluding LULUCF contributions. We derive target emissions estimates excluding LULUCF.

In order to quantify mitigation targets excluding LULUCF, and treat the LULUCF component separately, we classified target information from the NDCs into including and excluding LULUCF (inclLU and exclLU). In principle, when LULUCF is assessed to be covered and the NDC does not indicate it otherwise, the target information is assigned to inclLU (e.g., 20% reduction vs. BAU with LULUCF being covered is "RBU inclLU"), else to exclLU ("RBU exclLU"). Unfortunately, as it is not always clear, whether the NDC includes LULUCF in its mitigation target, and whether LULUCF emissions are included in provided baseline emissions, the classification sometimes relies on our judgement.

Target emissions are generally calculated based on Eq. 1 to 7 (Sect. 2.1.2), and we derive estimates both for $\mathrm{emiTarget_{inclLU}}$ and $\mathrm{emiTarget_{exclLU}}$. The emissions from LULUCF are treated separately when possible, but this is not always feasible. When, e.g., the quantification is based on NDC-data, and information on the LULUCF emissions contribution is not provided, no distinction is made between a LULUCF and a non-LULUCF part. If enough data are available, however, we use the





following approach to derive $\text{emiTarget}_{\text{inclLU}}$ and $\text{emiTarget}_{\text{exclLU}}$ (Table 8: example for India's REI_RBY target inclLU with LULUCF sink in the base year is assessed).

**Target excludes LULUCF**

– $\text{emiTarget}_{\text{exclLU}}$: use the given target emissions (ABS), or calculate them following Eq. 1 to 7 (LULUCF not considered in these equations).

    – Calculate $\text{emiTarget}_{\text{inclLU}}$ by adding the projected LULUCF emissions (no reduction of the LULUCF emissions):
$\text{emiTarget}_{\text{inclLU}} = \text{emiTarget}_{\text{exclLU}} + \text{emiBL}_{\text{onlyLU}_{\text{tarYr}}}$.

**Target includes LULUCF**

– Target types ABS, AEI, and ABU: use the ABS target as $\text{emiTarget}_{\text{inclLU}}$, or calculate $\text{emiTarget}_{\text{inclLU}}$ from AEI (multiplication with $\text{IntensityRef}_{\text{tarYr}}$), or from ABU (reduction of the BAU emissions in the target year by the given absolute reduction).

    – Target types RBY, REI, and RBU:

       – We assume the same mitigation effort in all sectors and apply the same relative reduction to all sectors, unless
stated differently in the NDC.

       – $\text{emiBL}_{\text{onlyLU}_{\text{refYr}}} > 0$ (net source): LULUCF treated as the other covered sectors and reduced by given relative reduction.

       – $\text{emiBL}_{\text{onlyLU}_{\text{refYr}}} < 0$ (net sink): sink is left as is. We chose not to strengthen the sink (attention if chosing to strengthen the sink: applying a relative reduction to negative values would weaken the sink potential). LULUCF
emissions and targets are connected to uncertainties (Fyson and Jeffery, 2019). Further, stringent non-LULUCF emissions reductions are of major importance for climate neutrality (IPCC, 2018a), and carbon sequestration in vegetation and soils comes with a time component (saturation of mitigation potential, created enhanced carbon stocks are reversible and non-permanent, Smith et al., 2014; vegetation or tree age, Pugh et al., 2019, Köhl et al., 2017, Stephenson et al., 2014, Carey et al., 2001).

– Calculate $\text{emiTarget}_{\text{exclLU}}$ by subtracting the projected LULUCF emissions:
$\text{emiTarget}_{\text{exclLU}} = \text{emiTarget}_{\text{inclLU}} - \text{emiBL}_{\text{onlyLU}_{\text{tarYr}}}$.

    – If for a country a resulting $\text{emiTarget}_{\text{exclLU}}$ becomes negative, which could only be achieved with negative emissions technologies and reliable sequestration, we use a *"second approach for LULUCF"*:

       – Split the absolute reduction in the target year against the baseline emissions $\text{ABU}_{\text{inclLU}}$ into $\text{ABU}_{\text{exclLU}}$ and
$\text{ABU}_{\text{onlyLU}}$, depending on the respective contributions of $\text{emiBL}_{\text{onlyLU}_{\text{tarYr}}}$ and $\text{emiBL}_{\text{exclLU}_{\text{tarYr}}}$
$(\text{ABU}_{\text{exclLU}} = (\text{ABS}_{\text{inclLU}} - \text{emiBL}_{\text{inclLU}_{\text{tarYr}}}) \cdot \frac{\text{emiBL}_{\text{exclLU}_{\text{tarYr}}}}{\text{emiBL}_{\text{inclLU}_{\text{tarYr}}}})$.

       – Reduce the baseline emissions $\text{emiBL}_{\text{exclLU}_{\text{tarYr}}}$ by the corresponding $\text{ABU}_{\text{exclLU}}$.

    – ABU targets: if the absolute reduction exceeds the assumed BAU emissions $\text{emiBL}_{\text{exclLU}_{\text{tarYr}}}$, the then negative target is set to $\text{emiTarget}_{\text{exclLU}} = 0 \text{ Mt CO}_2\text{eq}$.





For several countries, the Climate Action Tracker uses a somewhat comparable approach to derive target emissions excluding LULUCF from mitigation targets that include LULUCF – given in the NDC or calculated by applying the given reductions to the reference year emissions that include LULUCF: the projected LULUCF target year emissions are subtracted from the target emissions that include LULUCF (see, e.g., CAT, 2019a, b, for Australia and Brazil).

When quantifying all available targets, based on the *down-scaled marker scenario for SSP2 (dmSSP2)*, with NDC emissions
data prioritised if available, and assumed coverage of 100%, the *"second approach for LULUCF"* is needed for seven countries, and for Tonga the $\mathrm{ABU_{exclLU}}$ exceeds the baseline emissions $\mathrm{emiBL_{exclLU_{tarYr}}}$ (type_main: NGT, type_reclass: ABU).

### 2.4.2 Emissions pathways

One of our main goals is to construct global emissions pathways up to 2030, consistent with the NDC mitigation targets. For the aggregation, rather than quantified target emissions for single years, time series are needed, defined by interpolation between
target years and extrapolation after the last target year if it is before 2030. Pathway calculations start in 2021, the first year after the Kyoto Protocol period and the first year of the PA period (before 2021: baseline emissions), and a linear in- or decrease of the relative difference to the baseline is assumed between target years, while the relative difference is kept constant after the last target year (Table 5). If the baseline increases, a constant relative difference results in an increasing mitigation pathway, but with a smaller growth rate. To prevent the pathway from increasing a lot, the inter-annual baseline growth-rates are used if
the target in the last target year is above baseline. A second option for the calculation of national pathways is implemented in NDCmitiQ: constant emissions after the last target year (not default). For countries that indicated an emissions peak year, the calculated pathway is used in case it declines starting in the peak year or earlier, else the intended trajectory is approximated by keeping the national emissions constant after the peak year. Emissions baselines currently available in NDCmitiQ are either the constructed NDC emissions pathways (Sect. 2.2.3), or the down-scaled SSP marker scenarios (Sect. 2.2.1).

**Table 5.** Options for emissions pathway calculations for countries with a mitigation target but without a target for the year 2030. In this example, the country targets for a 20% reduction compared to BAU in 2025. The relative difference to the baseline emissions from 2020 to 2025 evolve linearly from 0% to -20%. After 2025, either the relative difference to the baseline is kept at the level of the last target year (default: option "constant percentages"), or the absolute emissions are kept at the level of the last target year (option "constant emissions"). The baseline emissions follow the chosen baseline scenario.

| Year | | **2020** | 2021 | 2022 | 2023 | 2024 | **2025** | 2026 | 2027 | 2028 | 2029 | 2030 |
|---|---|---|---|---|---|---|---|---|---|---|---|---|
| Baseline | Mt $CO_2$eq | 10.0 | 12.0 | 15.0 | 18.0 | 20.0 | 22.0 | 24.0 | 26.0 | 27.0 | 28.0 | 29.0 |
| *Constant* | % | **0.0** | -4.0 | -8.0 | -12.0 | -16.0 | **-20.0** | *-20.0* | *-20.0* | *-20.0* | *-20.0* | *-20.0* |
| *percentages* | Mt $CO_2$eq | 10.0 | 11.5 | 13.8 | 15.8 | 16.8 | 17.6 | 19.2 | 20.8 | 21.6 | 22.4 | 23.2 |
| *Constant* | % | **0.0** | -4.0 | -8.0 | -12.0 | -16.0 | **-20.0** | -26.7 | -32.3 | -34.8 | -37.1 | -39.3 |
| *emissions* | Mt $CO_2$eq | 10.0 | 11.5 | 13.8 | 15.8 | 16.8 | 17.6 | *17.6* | *17.6* | *17.6* | *17.6* | *17.6* |

We aim for globally aggregated emissions pathways per conditionality and range (in decreasing ambition order: unconditional best & worst, conditional best & worst). If for the current target type, conditionality, and range, values for emiTarget are





available for several years, all are used to construct the current pathway. Additionally, if an unconditional but no conditional target is stated for a certain year, we consider the unconditional target for the conditional pathway as well (if a target is available for X in year 20xx, but not for Y, also use X for the Y pathway; with [X & Y] in this order: [uncond. best & uncond. worst]; [cond. best & cond. worst]; [uncond. best & cond. best and worst]).

For countries without quantifiable mitigation target the baseline emissions are assumed as un / conditional pathways. Furthermore, if a country only has conditional targets, the baseline is used as unconditional pathways. However, in some of these cases the conditional pathway is worse than the baseline, which would result in a worse conditional than unconditional pathway. As this does not seem logical, the conditional (worst) pathway is also used as unconditional pathways if this happens. An option to disable this method and use the baseline as unconditional pathways nevertheless is implemented in the tool (not default).

The national pathways are finally aggregated to regional / global emissions pathways, per conditionality and range. Per country, one target type is prioritised for the aggregation, which can be type_main or type_reclass (Sect. 2.1.1). Further options to modify the target or pathway calculations are implemented in NDCmitiQ. These non-default options that can be chosen for comparison runs and sensitivity analyses are presented in the Sect. A4 and consist of the following options: "targets only for countries X, Y, Z", "prioritised target types", "countries without unconditional targets & what if baseline is better than the conditional targets", "set coverage to 100%", and "strengthen targets".

## 3 NDCmitiQ: examplary use cases

Throughout Sect. 2, the methodology of NDCmitiQ to assess NDCs and quantify their mitigation targets was explained, providing information on the data sets of emissions, population, and GDP currently in use in NDCmitiQ. We presented important background information needed for target calculations, and gave some insights on possible uncertainties. Now, we wish to demonstrate example use cases of the input and output data of NDCmitiQ: assessment of the covered share of emissions; baseline emissions from within NDCs compared to SSP baselines; national GHG mitigation targets: example India, with general importance of the results; and global mitigation pathways: influence of different quantification options.

### 3.1 Share of total emissions covered by NDC

On country-level we retrieved information on the sectors and gases covered (Fig. A1), and estimated the corresponding covered share of emissions (Table 6 and Fig. 5; excl. LULUCF). The Energy sector and the GHGs $CO_2$, $CH_4$, and $N_2O$ are considered in many NDCs (mentioned by 193, 174, 157, and 147 countries, respectively; countries that are part of the EU target counted as single countries). Globally, the Energy sector was responsible for 74.1% of emissions in 2017 (all shares in this section are for 2017 and excl. LULUCF and bunkers fuels), and a total of 74.5 / 16.9 / 6.5% was emitted in the form of $CO_2$ / $CH_4$ / $N_2O$. F-gases and the sector IPPU, representing 2.0 and 9.3% of global emissions, are least covered by NDCs, while F-gases have long atmospheric lifetimes and very high GWPs (e.g., AR4 GWPs: HFCs 4–14 800, PFCs 7 190–12 200). On a global scale, the Energy and IPPU sectors are dominated by $CO_2$ emissions, while Agriculture and Waste are dominated by $CH_4$ emissions.





On the country-level, shares vary, and in many African countries the highest emitting sector is Agriculture and the gas with the
highest share of national emissions is CH$_4$. Based on the available data for F-gases, they contributed 2.1% to global emissions
in 2017, with the majority of F-gases emitted in the form of HFCs (88.8%), followed by SF$_6$, PFCs, and NF$_3$ (5.6, 5.3 and 0.4%,
respectively). Their shares can be underestimated, however, as especially for NF$_3$, data are only available for few countries.
NF$_3$ was included in the Kyoto GHG basket in 2012 only (Doha amendment to the Kyoto Protocol, UNFCCC, 2012).

In total, we assess 77% of 2017' global emissions to be emissions from sectors and gases covered by the current NDCs. An
estimated 1% was emitted by countries without an NDC, plus about 14% by the USA. This leaves 9% of not-covered emissions
from countries with an NDC. Including the USA would increase the covered share significantly, to 91%. Article 4.4 of the PA
(UNFCCC, 2015) asks developed countries to implement economy-wide absolute emission reductions, which is reflected in
the high %cov for developed countries. 53 / 75 countries are assessed to cover less than 90 / 99% of their emissions, including
China and India, which contributed 27% and 6% of 2017' emissions, and for 43 countries emissions from the not-covered
sectors and gases gained in importance over recent years (negative trend of %cov). The influence of %cov on India's target
emissions, and on global scale, is further discussed in Sect. 3.3 and 3.4.

We do not consider the covered share of emissions for ABS and AEI targets, which can introduce an uncertainty. While the
99 countries classified as ABS and AEI targets for type_reclass (absolute emissions or absolute emissions intensity; excl. USA)
are responsible for one-fourth of global emissions (2017: 24.9%, 2030: 25.2% following dmSSP2; excl. LULUCF and bunkers
fuels), the not-covered share of emissions for these countries is only 0.4% of 2017' global emissions and the uncertainty
introduced is low.

### 3.2   Emissions data from the NDCs vs. dmSSPs

We retrieved emissions data from all NDCs with available data, and classified them as including or excluding LULUCF to
use the emissions in the target quantifications. In Table 7 the emissions from NDCs are compared with external baseline
emissions data (before 2017: national emissions from PRIMAP-hist v2.1 HISTCR (exclLU); after 2017: down-scaled SSP
marker scenarios (dmSSPs, exclLU); and LULUCF emissions data as described in Sect. 2.2). In all the historical years in
Table 7, the aggregated NDC baseline emissions are lower than the comparison baselines. To some degree, lower values can
be connected to a discrepancy between the sectors and gases that are included in the provided data, which are not always
clearly stated in the NDCs (comparison data: national totals). Even though the estimated baseline emissions for 2025 under
the NDCs are in the range of the dmSSPs, they are at the very upper edge – dmSSP5 – which is the most extreme pathway
with strongest emissions increase. For 2030, the aggregated NDC baseline emissions are even higher than dmSSP5, with
data from 25 / 42 countries available for the assessment of emissions exclLU / inclLU, and the countries representing 5.4% /
9.8% of global emissions in 2030 under dmSSP2 (middle-of-the-road scenario). The emissions estimates provided in NDCs
for 2030 are +34.8% / +97.5% or +1.3 / +6.1 Gt CO$_2$eq higher than dmSSP2 for the corresponding countries (for exclLU /
inclLU). Targets with reductions relative to Business-as-Usual emissions are higher, the higher the expected BAU emissions.
If an unrealistically strong increase in BAU emissions is assumed, it results in higher and easier to reach target emissions.
Another incentive for countries to have high baselines is that they can reflect a strongly growing economy. Using independent,





**Table 6.** Absolute and relative contribution of different gases and sectors to the global 2017 Kyoto GHG emissions (part a), and share of emissions covered by NDCs (part b). All values exclude emissions from LULUCF and bunkers fuels emissions. All values are based on PRIMAP-hist v2.1 HISTCR emissions data (GWP AR4). **(a)** Global emissions per sector and gas ("Emissions", in Gt $CO_2$eq). Remaining cells: global share per sector / gas (e.g., Energy contributed 74.1% to global 2017 emissions, and Energy $CO_2$ 67.3%). **(b)** Covered sectors / gases and corresponding emissions shares. "NDCs (Adapt.)": number of countries that stated (more or less explicitly) that they are covering a certain sector or gas (in brackets: adapted value based on above-given rules; EU: counting single countries). The given shares represent the part of emissions per sector plus gas combination that is estimated to be covered (relative to the global emissions from this sector-gas combination), and the total per sector or gas ("Share"; e.g., an estimated 80.7% / 82.8% of global Energy / Energy $CO_2$ emissions are covered). Countries with NGT targets that state covered sectors and gases are included in the presented numbers. Complementary information is provided in Table A5.

**(a) Absolute and relative contributions to global emissions, per gas & sector**

| 2017 | Emissions | Share | CO$_2$ | CH$_4$ | N$_2$O | HFCs | PFCs | SF$_6$ | NF$_3$ |
|---|---|---|---|---|---|---|---|---|---|
| *Emissions* | 47.7 Gt CO$_2$eq | | 35.5 | 8.1 | 3.1 | 0.9 | 0.1 | 0.1 | 0.0 |
| *Share* | | 100.0% | 74.5% | 16.9% | 6.5% | 1.8% | 0.1% | 0.1% | 0.0% |
| **Energy** | 35.3 | **74.1%** | 67.3% | 6.3% | 0.6% | – | – | – | – |
| **IPPU** | 4.4 | **9.3%** | 6.8% | 0.0% | 0.4% | 1.8% | 0.1% | 0.1% | 0.0% |
| **Agriculture** | 6.1 | **12.8%** | 0.3% | 7.5% | 4.9% | – | – | – | – |
| **Waste** | 1.6 | **3.4%** | 0.1% | 3.1% | 0.3% | – | – | – | – |
| **Other** | 0.2 | **0.4%** | 0.0% | 0.0% | 0.4% | – | – | – | – |

**(b) Gases and sectors covered by NDCs**

| 2017 | NDCs (Adapt.) | Share | CO$_2$ | CH$_4$ | N$_2$O | HFCs | PFCs | SF$_6$ | NF$_3$ |
|---|---|---|---|---|---|---|---|---|---|
| *NDCs (Adapt.)* | | | 174 (193) | 157 (175) | 147 (165) | 78 (79) | 71 (72) | 71 (73) | 51 (53) |
| *Share* | | 76.9% | 83.5% | 61.3% | 55.3% | 33.5% | 36.7% | 46.1% | 15.8% |
| **Energy** | 193 (193) | **80.7%** | 82.8% | 60.5% | 53.4% | – | – | – | – |
| **IPPU** | 123 (142) | **75.3%** | 90.1% | 58.0% | 35.4% | 33.5% | 36.7% | 46.1% | 15.8% |
| **Agriculture** | 139 (143) | **57.5%** | 75.3% | 56.7% | 57.6% | – | – | – | – |
| **Waste** | 149 (148) | **73.9%** | 99.7% | 74.1% | 66.8% | – | – | – | – |
| **Other** | 0 (124) | **40.0%** | 100.0% | 100.0% | 40.0% | – | – | – | – |



**Figure 5.** (a & b) highest & second highest contributing sector plus gas combination on a national scale (2017 emissions). (c) global share of national emissions for 2017 (non-linear colour scale). (d) average emissions trend 2010–2017 in %/year (based on linear regression to national emissions 2010–2017). (e) share of Kyoto GHG emissions assumed to be covered by a country's NDC mitigation target (for 2017). (f) average trend of %cov 2010–2017 in %/year (based on linear regression to national %cov 2010–2017). All values are based on PRIMAP-hist v2.1 HISTCR emissions, following GWP AR4 and exclude emissions from LULUCF and bunkers fuels. USA: information from NDC shown here for information purposes only.





country-specific comparison data is helpful to set national estimates into perspective. However, for the purpose of quantifying the NDCs' mitigation targets, it is most helpful to use the BAU emissions provided in the Parties' documents, if available, as

this is most consistent with what the country has pledged.

**Table 7.** Baseline emissions data provided in NDCs compared to our baseline emissions (separated into excluding & including emissions from LULUCF). For the base and target years of current mitigation targets, all emissions data provided in the NDCs are aggregated (row "NDCs"), and compared to our baseline emissions (aggregate over the same countries; rows "dmSSP1–5" with same values for 1990–2017). Baseline emissions: see Sect. 2.2 (PRIMAP-hist v2.1 1990–2017, dmSSPs, LULUCF emissions, all excl. bunkers fuels). "Difference to dmSSP2": how do the NDC values compare to the dmSSP2 baseline; "Number of Parties": the number of Parties with emissions data available; "Global share (dmSSP2)": the global share of emissions for the countries with data available in a certain year, compared to dmSSP2 (excl. bunkers fuels). NDC emissions based on the GWP from SAR were converted to AR4 using national conversion factors.

| (*): Mt $CO_2$eq AR4 | 1990 | 2000 | 2005 | 2006 | 2010 | 2013 | 2014 | 2025 | 2030 |
|---|---|---|---|---|---|---|---|---|---|
| **Excluding LULUCF** | | | | | | | | | |
| dmSSP1 (*) | | | | | | | | 140.1 | 3 394.2 |
| dmSSP2 (*) | | | | | | | | 144.0 | 3 590.9 |
| dmSSP3 (*) | 1 353.1 | 0.2 | 293.0 | 0.1 | 27.4 | 1 412.2 | 0.2 | 154.2 | 3 996.7 |
| dmSSP4 (*) | | | | | | | | 149.3 | 3 808.8 |
| dmSSP5 (*) | | | | | | | | 159.6 | 4 158.4 |
| NDCs (*) | 1 318.7 | 0.1 | 273.6 | 0.1 | 8.5 | 1 408.0 | 0.2 | 158.4 | 4 841.8 |
| Difference to dmSSP2 | -2.5% | -20.1% | -6.6% | -16.1% | -69.0% | -0.3% | -22.9% | +10.0% | +34.8% |
| Number of Parties | 7 | 1 | 1 | 1 | 3 | 1 | 1 | 8 | 25 |
| Global share (dmSSP2) | 4.2% | 0.0% | 0.7% | 0.0% | 0.1% | 3.1% | 0.0% | 0.2% | 5.4% |
| **Including LULUCF** | | | | | | | | | |
| dmSSP1 (*) | | | | | | | | 540.4 | 5 816.7 |
| dmSSP2 (*) | | | | | | | | 550.0 | 6 227.8 |
| dmSSP3 (*) | 1 063.4 | | 3 612.5 | | 39.4 | | | 596.3 | 6 947.8 |
| dmSSP4 (*) | | | | | | | | 572.5 | 6 541.7 |
| dmSSP5 (*) | | | | | | | | 625.2 | 7 515.0 |
| NDCs (*) | 1 011.8 | | 3 147.5 | | 28.6 | | | 624.8 | 12 301.3 |
| Difference to dmSSP2 | -4.9% | | -12.9% | | -27.4% | | | +13.6% | +97.5% |
| Number of Parties | 6 | | 4 | | 2 | | | 12 | 42 |
| Global share (dmSSP2) | 3.3% | | 9.1% | | 0.1% | | | 1.0% | 9.8% |

### 3.3 India's emissions intensity target: quantification and challenges

As an example of national target quantifications with NDCmitiQ, we present an analysis of parts of India's NDC. We show India as an example because several points made below are not specific to India's NDC, but of general interest and concern.

In its NDC, India presents a GHG mitigation target of a 33–35% reduction in emissions intensity per unit of GDP, with the chosen base and target year being 2005 and 2030 respectively (Republic of India, 2016). As India has only reported emissions data to the UNFCCC for 1994, 2000, and 2010, no data were reported for the chosen base year 2005, and the 1994 data were reported before the 1996 IPCC guidelines for national GHG inventories (IPCC, 1996) were introduced. While developed countries (Annex-I Parties) are obliged to submit annual GHG inventories to the UNFCCC, India, as a developing country, is not. Under the Katowice Climate Package (UNFCCC, 2019), however, with self-determined flexibility, "Each Party shall report a consistent annual time series starting from 1990; those developing country Parties that need flexibility in the light of their capacities with respect to this provision have the flexibility to instead report data covering, at a minimum, the reference year / period for its NDC under Article 4 of the Paris Agreement and, in addition, a consistent annual time series from at least 2020 onwards" (II.E.3.57).

India does not clearly state the covered gases and sectors, but rather gives measures for different sectors. We assessed the covered sectors to be Energy, IPPU, Waste, and LULUCF, and as no information is provided on the considered gases, $CO_2$, $CH_4$, and $N_2O$ are assumed to be covered, resulting in an estimated %cov of 74 / 86% in 2005 / 2030 (excl. LULUCF; compared to the dmSSP2 baseline emissions). India's total emissions in 2017 were 6.3% of global Kyoto GHG emissions (excl. LULUCF and bunkers fuels; based on PRIMAP-hist v2.1 HISTCR AR4). Figure 6 shows the importance of the covered $CO_2$ emissions from the Energy sector (71% of India's emissions in 2017), and the steady and steep increase over recent years. In the fiscal year that ended in March 2020, India's $CO_2$ emissions fell by more than 1%, for the first time in almost four decades, and this decline is not only caused by the Covid-19 lock-down, as it already started in early 2019 (Myllyvirta and Dahiya, 2020).

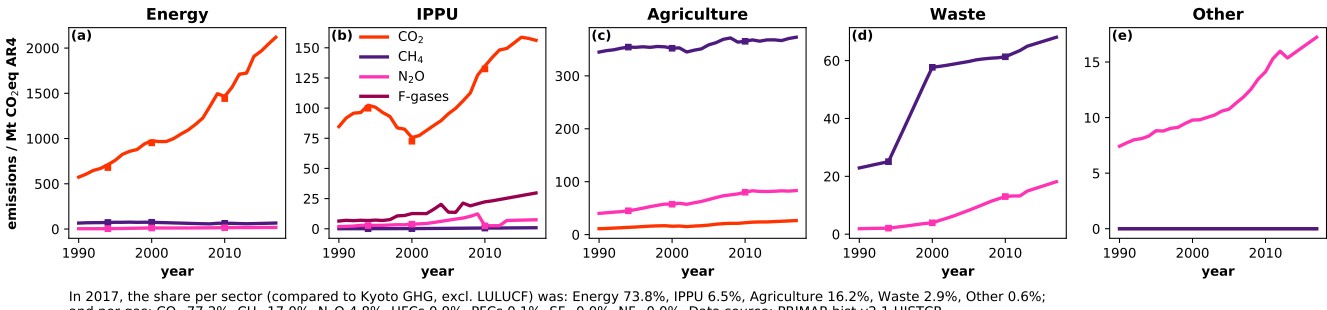

In 2017, the share per sector (compared to Kyoto GHG, excl. LULUCF) was: Energy 73.8%, IPPU 6.5%, Agriculture 16.2%, Waste 2.9%, Other 0.6%; and per gas: $CO_2$ 77.2%, $CH_4$ 17.0%, $N_2O$ 4.8%, HFCs 0.9%, PFCs 0.1%, $SF_6$ 0.0%, $NF_3$ 0.0%. Data source: PRIMAP-hist v2.1 HISTCR.

**Figure 6.** India's historical emissions 1990–2017. Panel (a) to (e): emissions per main-sector, split into the contributing Kyoto GHGs ($CO_2$, $CH_4$, $N_2O$, and in the case of IPPU additionally F-gases as a total). Additionally, the share of per-sector and per-gas emissions in 2017 is presented, compared to the national totals (Kyoto GHG excl. LULUCF; as text below the figure). Please note the different vertical axes limits. Data source: PRIMAP-hist v2.1 HISTCR. The raw country-reported data (UNFCCC 2019) are additionally presented as squares (no data available for the different F-gases).

The target quantifications are based on the external data described in Sect. 2.2, as, besides an estimate of 2030 GDP, we did not find the necessary data in India's NDC. Together with the corresponding baseline emissions and GDP scenarios,





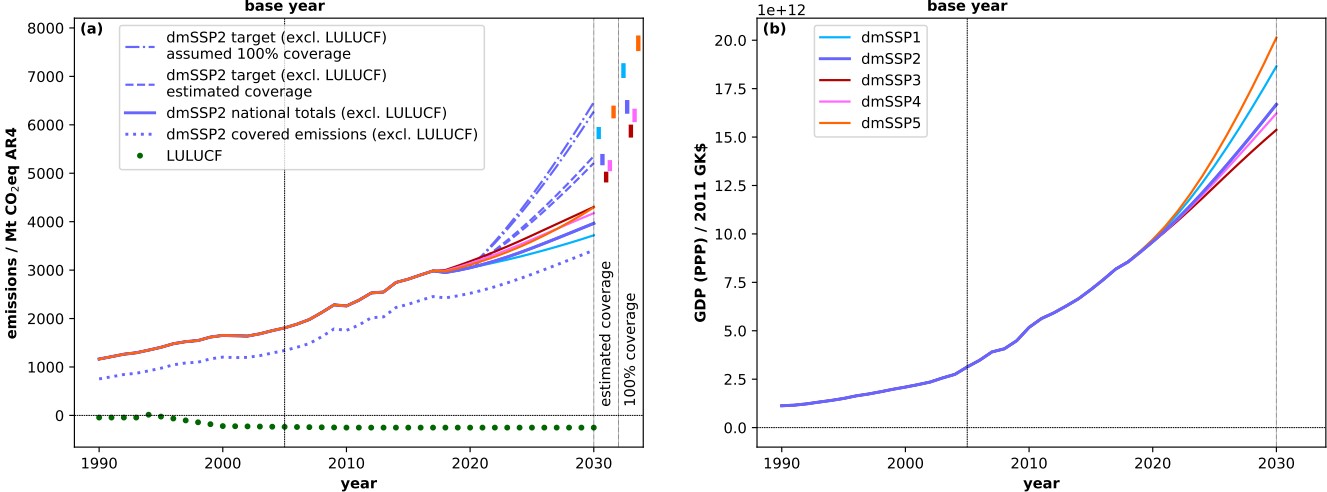

**Figure 7.** Emissions (a) and GDP (b) time series for India. (a) emissions from the down-scaled SSP2 marker scenario (solid blue line), the corresponding covered share of emissions (dotted blue line), and LULUCF emissions (dotted green line, UNFCCC 2019; inter- and extrapolated). NDC GHG mitigation target emissions (33–35% reduction in emissions intensity per unit of GDP in 2030, compared to 2005) are shown for 2030 per SSP marker scenario. Quantifications based on an estimated coverage of 74 / 86% in 2005 / 2030: "estimated coverage". Based on an assumed coverage of 100%: "100% coverage". All emissions exclude LULUCF (besides "LULUCF"). (b) GDP time series for the SSP marker scenarios (unit 2011 GK$: 2011 Geary-Khamis international dollars).

quantifications based on dmSSP1–5 are compared, once for an assumed 100% coverage, and once based on the estimated
%cov (Fig. 7). There are at least three interesting aspects:

**(i)** The 2030 mitigation targets lie above the baseline emissions for all dmSSPs, mainly caused by the projected growth in GDP. India would overachieve the intensity target if the assumed baseline emissions were met, and there seems to be room for a more ambitious target than a 33–35% reduction in emissions intensity per unit of GDP. The GDP-based downscaling of regional SSP emissions scenarios suggests that the targets could be more stringent. For the middle-of-
the-road scenario dmSSP2, India's GDP is assumed to increase by a factor of approximately 5.3 from 2005 to 2030. India provides an estimate of its 2014 and 2030 GDP at 2011-12 prices (in trillion): 1.69 and 6.31 US$ (Republic of India, 2016). This would constitute an increase by a factor of 3.7 from 2014 to 2030, and with linear approximation a 5.8 times raise from 2005 to 2030. Assumed linearity probably leads to an overestimation, and the factor is in line with the GDP growth factor of 5.3 from dmSSP2. The assumed baseline emissions also affect these findings, as, if we would assume
significantly higher baseline emissions in 2030 than presented, while not changing any of the remaining assumptions, the target emissions would no longer be above the increased baseline emissions.

**(ii)** For the different dmSSPs, the order of targets from highest to lowest is dominated by the GDP growth factor, and not by the increase in baseline emissions (more details in Sect. 3.4)





**(iii)** The targets with assumed 100% coverage are higher than with estimated coverage of 74% in 2005 and 86% in 2030
(details below).

The unexpected behaviour of the targets with an assumed coverage of 100% being higher than the comparison with estimated %cov is, in a mathematical sense, a combination of two aspects: (i) the high projected GDP growth rate, and (ii) the increase in the share of covered base year emissions (example for dmSSP2: equations and estimates in Table 8). When %cov increases, $\mathrm{emiBL_{COV_{2005}}}$ and therefore the first term of the equation for $\mathrm{emiTarget_{inclLU}}$ increases, while the last term
($\mathrm{emiBL_{notCOV_{2030}}}$) decreases and reaches 0 Gt $CO_2$eq for 100% coverage. For India's target, the rise in the first term is not compensated for by the decline of the last term of the equation, leading to the observed higher target emissions for 100% coverage. However, several aspects would work against this behaviour. If the projected GDP growth rate was significantly lower or the down-scaled 2030 baseline emissions were significantly higher (GDP growth factor below 1.7,or reference emissions higher than 12 Gt $CO_2$eq in this example), the behaviour would not occur and moving towards 100% coverage would result
in lower target emissions that would lie below the 2030 baseline (REI_RBY with growth factor of 1: same as RBY target). Furthermore, and importantly, if the target value (relative reduction in emissions intensity per unit of GDP) itself was strong enough, and not weaker than the baseline assumptions, this behaviour would not occur, and at the same time the target emissions would not exceed baseline emissions (with numbers as in Table 8: with a 53% / 59% reduction the target with estimated / 100% coverage, respectively, would be below baseline; and with a 78% reduction the 100%-coverage target would be below
the estimated-coverage target). No information on the part of national GDP corresponding to the different emissions sectors is included in the assessment of the covered share of emissions. Doing so can change the results, and nations should consider the emissions intensities of added sectors when updating targets to expand the scope of the pledges.

     The results should not be misunderstood as a motivation not to move towards an economy-wide target and including all Kyoto GHGs and sectors in the mitigation target, as aimed for by the PA. Our findings rather show that while doing so, in some
cases Parties need to assess whether they have to increase their reduction level simultaneously, or move to a different target type overall, to ensure to ramp up the ambition rather than to lower it, and point towards quantification challenges and target uncertainties. For a few other countries our results also show higher target emissions when shifting towards a 100% coverage, compared to the estimated coverage. The countries for which this happens for all five dmSSPs, are India (REI), Uzbekistan (REI), Botswana (RBY), the Democratic Republic of the Congo (RBU), and Tajikistan (RBY). China's target (REI) is also
higher for a 100% coverage, but only for dmSSP1 and dmSSP5, the scenarios with highest projected GDP growth, and smallest growth factor for national emissions per unit of GDP.

     The coverage for India's mitigation target is prone to uncertainty, as it is not clearly communicated in the NDC and leaves room for interpretation. Based on India's NDC, we did not assess the Agriculture sector to be covered. The CAT (2019d) also assumes the Agriculture sector to be excluded, based on the information on the 2020 pledges, "even though not mentioned
in the NDC", and Climate Watch (a) and the World Resources Institute state the "Sectors covered" as "Not specified; various sectors mentioned for mitigation and adaptation strategies such as energy, industry, transportation, agriculture, forestry, waste". Consistent with our assessment of India's NDC, the NDC Explorer (Pauw et al., 2016) states "Not indicated" for "Mitigation





**Table 8.** Quantification of India's emissions intensity target (REI). Input data for dmSSP2 and target equations ($\text{emiTarget}_{\text{inclLU}}$ and $\text{emiTarget}_{\text{exclLU}}$), for 100% coverage and estimated coverage (2005 / 2030: 74 / 86%). LULUCF baseline emissions in 2005 negative, therefore $\text{emiBL}_{\text{onlyLU}_{2005}}$ is not reduced or strengthened. Presented values contain rounding artefacts (results based on values with higher precision than the shown input data).

| **Unconditional worst target** | | | **GDP growth factor** | |
|---|---|---|---|---|
| 33% reduction: $\text{NDC}_{\%\text{level}} = 100\% - 33\% = 67\%$ | | | $\frac{\text{GDP}_{2030}}{\text{GDP}_{2005}} = \frac{1.7\cdot10^{13}\ 2011\text{GK\$}}{3.1\cdot10^{12}\ 2011\text{GK\$}} = 5.3$ | |
| **Baseline emissions (exclLU)** | | | **Baseline emissions (onlyLU)** | |
| $\text{emiBL}_{\text{exclLU}_{2005}}$ | $\text{emiBL}_{\text{exclLU}_{2030}}$ | | $\text{emiBL}_{\text{onlyLU}_{2005}}$ | $\text{emiBL}_{\text{onlyLU}_{2030}}$ |
| 1.8 Gt CO$_2$eq | 4.0 Gt CO$_2$eq | | $-0.2$ Gt CO$_2$eq | $-0.3$ Gt CO$_2$eq |

$$\textbf{emiTarget}_{\textbf{inclLU}} = [\, \text{NDC}_{\%\text{level}} \cdot \tfrac{\text{GDP}_{2030}}{\text{GDP}_{2005}} \cdot \text{emiBL}_{\text{COV}_{2005}} + \text{emiBL}_{\text{onlyLU}_{2005}} \,] + \text{emiBL}_{\text{notCOV}_{2030}}$$

Estimated coverage (74% in 2005, 86% in 2030):

| $[\,67\% \cdot 5.3 \cdot (\,74\% \cdot 1.8$ Gt CO$_2$eq$)$ | $+\ -0.2$ Gt CO$_2$eq $]$ | $+\ ((100\% - 86\%) \cdot 4.0$ Gt CO$_2$eq$)$ | $=$ |
|---|---|---|---|
| $[\,4.8$ Gt CO$_2$eq | $+\ -0.2$ Gt CO$_2$eq $]$ | $+\ 0.6$ Gt CO$_2$eq | $=\ 5.1$ Gt CO$_2$eq |

100% coverage (100% in 2005 and 2030):

| $[\,67\% \cdot 5.3 \cdot (100\% \cdot 1.8$ Gt CO$_2$eq$)$ | $+\ -0.2$ Gt CO$_2$eq $]$ | $+\ ((100\% - 100\%) \cdot 4.0$ Gt CO$_2$eq$)$ | $=$ |
|---|---|---|---|
| $[\,6.5$ Gt CO$_2$eq | $+\ -0.2$ Gt CO$_2$eq $]$ | $+\ 0.0$ Gt CO$_2$eq | $=\ 6.2$ Gt CO$_2$eq |

$$\textbf{emiTarget}_{\textbf{exclLU}} = \text{emiTarget}_{\text{inclLU}} - \text{emiBL}_{\text{onlyLU}_{2030}}$$

| Estimated coverage (74% in 2005, 86% in 2030): | 5.1 Gt CO$_2$eq $-\ -0.3$ Gt CO$_2$eq $=\ 5.4$ Gt CO$_2$eq |
|---|---|
| 100% coverage (100% in 2005 and 2030): | 6.2 Gt CO$_2$eq $-\ -0.3$ Gt CO$_2$eq $=\ 6.5$ Gt CO$_2$eq |

focus areas: agriculture", and for "Reducing non-CO$_2$ gases" it indicates "Considered (CH$_4$, N$_2$O)". As "GHG coverage", The World Bank (2016) states "n/a".

Another source of uncertainty is the conditionality of the target. India's NDC states "To mobilize domestic and new & additional funds from developed countries to implement the above mitigation and adaptation actions in view of the resource required and the resource gap" (Republic of India, 2016), and we classify it as unconditional, even though it is unclear to us if parts are conditional. Contrary to our assessment and the CAT (2019c), Climate Watch (a) and the World Resources Institute denote India's target as conditional.

Based on quantifications under dmSSP2 and an assumed 100% coverage, India's emissions target ranges between 6.3–6.5 Gt CO$_2$ eq for emissions excluding LULUCF (6.0–6.2 Gt CO$_2$ eq including LULUCF; AR4). With estimated coverage of 74 / 86% for 2005 / 2030, the quantified emissions target ranges between 5.2–5.4 Gt CO$_2$ eq for emissions excluding LULUCF (5.0–5.1Gt CO$_2$ eq including LULUCF). The CAT (2019c) estimates the unconditional emissions intensity target to be in the range of 6.0–6.2 Gt CO$_2$eq (excl. LULUCF, AR4). This value is a bit lower than our estimates when assuming a
100% coverage. Climate Watch (a) and the World Resources Institute give a wider range of 5.9–9.1 Gt CO$_2$eq, not specifying whether these emissions include or exclude LULUCF. The exact reasons for the quantification discrepancies could not be assessed, but chances are higher that differences arise from assumptions of projected than from historical data (LULUCF and





non-LULUCF emissions, GDP). In the short-term, India does not plan to raise the ambition of its NDC (Prakash Javadekar, minister of environment, forests and climate change: "The raising of ambition or ratcheting-up will arise only after a global
stocktake in 2023.", Gombar, 2020).

### 3.4   Global mitigation pathways

One of the main outputs of NDCmitiQ are global emissions pathways consistent with the NDC GHG mitigation targets. There-
fore, moving from example analyses of national targets and the underlying emissions data to global emissions, Fig. 8 shows
globally aggregated pathways resulting from a full implementation of current targets from unconditional worst to conditional
best, and based on different input data and quantification options. Once, the emissions data from the NDCs are prioritised
(type_reclass), and second the external time series are used (based on type_main). In the following, the mitigated emissions
pathways under the five SSPs are named "NDCSSP", while the baselines are named "dmSSP".

First, we analyse the impact of the targets' conditionality and different scenarios for emissions, population, and GDP on the
mitigation pathways. The higher aggregated emissions data from the NDCs for 2030 compared to the dmSSPs (Sect. 3.2) lead
to higher global baseline emissions (difference between "NDC and SSP baselines": dmSSP1–5 between 1.6–2.7 Gt $CO_2$eq
AR4), and consequently result in higher quantified mitigated emissions (NDCSSP1–5).

With our tool we confirm findings by Benveniste et al. (2018) that "the main sources of uncertainty is the range of ambitions
given in NDCs, and the uncertainty on the economic growth of countries who expressed their target in terms of intensity". In
the presented quantifications, the conditionality range is 2.8–6.0 Gt $CO_2$eq for all values displayed in Fig. 8 (panel b: difference
between unconditional worst and conditional best), but with little difference between the conditional worst and best emissions.
For the different dmSSP scenarios, we observe a strong influence of the projected GDP on the global mitigation results.
NDCSSP5 (fossil-fuelled development) has by far higher global emissions than NDCSSP1–4, which exceeds the difference
between the dmSSP1-5 baseline emissions, and results from the combination of high projected emissions baselines and GDP
growth. NDCSSP1–4 are approximately in the same range, with lowest mitigated emissions for NDCSSP2 (SSP2: often used
middle-of-the-road scenario) even though its emissions baseline is not the lowest, and NDCSSP3+4 have the highest quantified
mitigation impacts due to the lower GDP projections.

Only eight countries are assessed to have REI targets (relative reduction compared to emissions intensity), but amongst them
are China and India. The REI countries represented 16% of global emissions in 1990, but their share almost doubled by 2017
(35%), and is projected to further increase to 38% by 2030 (dmSSP2). Only the Dominican Republic chose its population
and not its GDP as emissions intensity reference. The influence of the underlying GDP data demonstrates the importance of
reasonable estimates of GDP to quantify the mitigation targets. The results are also in line with Rogelj et al. (2017), who found
the dominant driver of uncertainty in estimates of NDC mitigation levels on a global scale to be the potential variation in the
underlying socioeconomic assumptions.

The global aggregates for the mitigated pathways are generally below the corresponding baseline emissions scenarios. How-
ever, for NDCSSP1, with the lowest baseline emissions, but one of the highest GDP projections, this is not true (for uncon-
ditional worst). Higher mitigated than baseline emissions can result from all assessed target types excluding RBU and ABU.



**Figure 8.** (Panel a) Global baseline emissions for dmSSP1–5. Shaded areas: emissions pathways for dmSSP2 ("default" with estimated or assumed 100% coverage, for "prio NDCs" and "prio SSPs"). "(1) default": LULUCF data prioritisation CRF, BUR, UNFCCC, and then FAO & constant relative difference to baseline after last target year & cond. pathway used as uncond. pathway if the baseline is below the cond. pathway (and country has no uncond. target); "prio NDCs": prioritising NDC emissions data, based on type_reclass and the emissions described in Sect. 2.2.3; "prio SSPs": based on type_main and the dmSSPs. (Panel b) Estimates of mitigated emissions for 2030 based on the current NDCs. Vertical lines: range of un- / cond. best / worst targets (conditionality and range indicated by squares). Results based on the following options (altering one option per quantification): "(2) constant emi": constant emissions after last target year; "(3) baseline uncondi": baseline emissions used as uncond. pathway if country has no uncond. target; "(4) LU FAO": FAO as first rank of LULUCF source prioritisation. All emissions in panel a & b exclude LULUCF and bunkers fuels. (Panel c & d) Global pathways for the marker scenarios dmSSP1–5 baseline population and GDP.





Reductions compared to BAU emissions in the target year will be below baseline as long as the given NDC values are real reductions. Out of the presented runs, NDCSSP1 has the highest number of countries (23–29 countries for quantifications with 100% or estimated coverage, and type_main and type_reclass) for which the worst mitigated pathways are above the countries'
baseline emissions.

The effect of different assumptions of underlying LULUCF baseline emissions on the target quantifications on global scale are shown in Fig. 8. All emissions exclude LULUCF, but in many cases the targets exclLU have to be derived from targets inclLU (countries including LULUCF: Figure A1), and therefore the LULUCF baselines often affect exclLU targets. As default, LULUCF data from NDC, CRF, BUR, UNFCCC, and then FAO is prioritised. Prioritising FAO over CRF data leads to lower
target emissions on a global scale, even though the global LULUCF emissions estimate for 2030 is +3.4 Gt $CO_2$eq if FAO has the highest prioritisation, while it is a net sink of -2.2 Gt $CO_2$eq for prioritisation of CRF data. This behaviour is not connected to certain target types and we could not find a general pattern in the per-country changes that leads to this decrease in target emissions on a global scale. (Fyson and Jeffery, 2019) focused on the LULUCF component in NDCs, and studied uncertainties due to NDCs' LULUCF contributions in a more refined way. They found that the ambiguity in the emissions reductions due to
land-based activities results in ~3 Gt $CO_2$ / year uncertainty in 2030, which is larger than their estimated total anthropogenic land use sink of -2 Gt $CO_2$ / year in 2030, and larger than the influence our choice of underlying LULUCF data has on the quantified targets (0.8 Gt $CO_2$eq in global mitigated emissions exclLU for CRF vs. FAO).

To analyse their uncertainties, different options for the target and pathway calculations are implemented in NDCmitiQ. The effects of changing the options are smaller than the impact of conditionality and input data. For two options, the upper
limit of the range between unconditional worst and conditional best estimates is reduced, while the lower limit is unchanged: option (2) using the baseline emissions as uncond. pathway instead of the cond. pathway even if the baseline is lower than the cond. pathway (does not affect cond. pathways); or option (3) keeping the absolute emissions constant after a country's last target year instead of the relative difference to the baseline (only affects countries with last target year before 2030). What has been observed for India's target in Sect. 3.3 – higher target emissions for 100% coverage vs. estimated %cov – is seen on a
global scale as well, as India and China have high national emissions and are amongst the few countries for which the target quantifications show this behaviour (China: only for dmSSP1 and 5).

Depending on the quantification options and underlying dmSSP scenarios, global mitigated emissions under the NDCs in 2030 are estimated to range between 49.2 and 55.7 Gt $CO_2$eq for 2030 ($10^{th}$ / $90^{th}$ percentiles for unconditional worst and conditional best estimates for dmSSP1–4, with median 52.4 Gt $CO_2$eq; AR4; excl. LULUCF and bunkers fuels). Both, the
6.5 Gt $CO_2$eq range and absolute values are lower than the 56.8–66.5 Gt $CO_2$eq / year estimates by Benveniste et al. (2018) for 2030 (90% confidence interval; 9.7 Gt $CO_2$eq range; Table 9). However, adding to the difference is that their estimates include emissions from bunkers fuels, and probably LULUCF emissions, with "the share of international aviation and shipping in global emissions increas[ing] from 2.3% in 2010 to 3.0–3.7% in 2030" Benveniste et al. (2018). While they noted that essentially due to a range of GDP scenarios being considered instead of a single scenario the uncertainty range is larger than
previous studies, the smaller range of 6.6 Gt $CO_2$eq / year for the SSP2 OECD scenario is comparable to other estimates. With 4.1 Gt $CO_2$eq our median range for dmSSP2 is smaller. For 2030, the United Nations Environment Programme (2019)





found the global emissions for unconditional NDCs to be 56 Gt $CO_2$eq (54–60 Gt $CO_2$eq; median and $10^{th}$/$90^{th}$ percentiles; probably including LULUCF and bunkers fuels), and for conditional NDCs 54 Gt $CO_2$eq (51–56 Gt $CO_2$eq). Our estimates that exclude LULUCF emissions and bunkers fuels are 54.6 Gt $CO_2$eq for the upper edge (52.8–56.4 Gt $CO_2$eq, unconditional worst), and 50.4 Gt $CO_2$eq (48.9–51.6 Gt $CO_2$eq) for conditional best, representing a larger range.


**Table 9.** Comparison of mitigated global emissions for the year 2030 with United Nations Environment Programme (2019) and Benveniste et al. (2018). Benveniste et al. (2018) and their Supplementary Information:"share of international aviation and shipping in global emissions increase from 2.3% in 2010 to 3.0 to 3.7% in 2030"; "International aviation emissions for 2030 are approximated to lie within a range of 906 to 1200 Mt $CO_2$ $yr^{-1}$ [...]. International shipping emissions are based on projections [...] resulting in a range of emissions of 940 to 1200 Mt $CO_2$eq $yr^{-1}$ in 2030." United Nations Environment Programme (2019): "[...] with international transport (aviation and shipping) representing around 2.5 per cent of GHG emissions [in 2018, excluding LUC]".

|  | Results (Gt $CO_2$eq) | Information |
|---|---|---|
| Current study | 52.4 (49.2–55.7) | Median ($10^{th}$–$90^{th}$ percentiles), AR4, excl. LULUCF, excl. bunkers fuels. Based on quantifications with various input data (dmSSP1–4, prio NDCs, prio SSPs), 100% and estimated coverage, and additional options (see Fig. 8). |
|  | 54.6 (52.8–56.4) | Upper edge, unconditional NDCs. |
|  | 50.4 (48.9–51.6) | Lower edge, conditional NDCs. |
| United Nations Environment Programme (2019) | 56 (54–60) | Median ($10^{th}$–$90^{th}$ percentiles), unconditional NDCs, probably incl. LULUCF and bunkers fuels. |
|  | 54 (51–56) | Conditional NDCs. |
| Benveniste et al. (2018) | 61.7 (56.8–66.5) | 90% confidence interval (for all assessed scenarios), incl. LULUCF and bunkers fuels. Bunkers fuels in 2030: 2.4 Gt $CO_2$eq (calculated from emissions estimates provided in their study). |
|  | 61.8 (58.4–65.0) | For SSP2 OECD scenario. |

## 3.5 Other possible use cases

Additional use cases of NDCmitiQ and its output data include: climate change impact assessments based on the global emissions pathways; calculation of mid-century targets; analyses similar to Fig. 8, but on regional level, with refined view on target types, or changing several calculation options at a time to estimate interactions; effect of uncertainties in historical emissions;

comparisons with the allowable carbon budget for the PA temperature goals; and estimation of end-of-century temperature rise. To estimate the global temperatures for the year 2100 based on NDC mitigation pathways, in comparison with pre-industrial times, the aggregated emissions pathways can be used in combination with additional tools. The emissions time series can be extended to 2100 using the pathway extension by Gütschow et al. (2018) and the Kyoto GHG basket emissions can then be split into multi-gas pathways in the Equal Quantile Walk (Meinshausen et al., 2006). These multi-gas emissions pathways are





input needed to derive estimates of the probabilistic global mean temperatures by running the simple climate model "Model for the Assessment of Greenhouse Gas Induced Climate Change" (MAGICC; Meinshausen et al., 2011).

## 4 Discussion

This paper shows the methodology behind NDCmitiQ and possible use cases of this newly available open-source tool to quantify and analyse national GHG mitigation targets as stated in the current set of NDCs, and construct corresponding national
and global emissions pathways. NDCmitiQ is fast-running and incorporates a large amount of information retrieved from NDCs. It has a uniform approach with flexible input data for comparison studies, but also provides target quantifications based on the available emissions data in NDCs whenever possible. As the presented time series of emissions, population, and GDP data currently implemented in NDCmitiQ are not intended to be exclusive, users can add other suitable time series for the quantifications. We believe that NDCmitiQ can help researchers and stakeholders for fast analyses when updated NDCs are
submitted, or in the Global Stocktake.

The 168 NDCs assessed in our study, with documents consisting of three to 83 pages and strongly differing content and clarity, often leave room for interpretation. The "clarity, transparency, and understanding" (Art. 4.8; UNFCCC, 2015) of mitigation targets in NDCs could be improved substantially by, i.a., including estimates of the absolute target emissions; providing the underlying data; specifically specifying LULUCF emissions and targets in this sector; estimating the part of emissions
targeted by mitigation measures in the base and target year, and providing information on what is expected to happen with the emissions from not-covered sectors and gases; giving information on the planned evolution of emissions after the last target year. Implementation of the Katowice Climate Package (UNFCCC, 2019) will hopefully reduce some of these sources of quantification uncertainties. However, as the above mentioned clarity is still missing in the current set of NDCs, we addressed the corresponding challenges and uncertainties throughout this manuscript and provide possible quantification options
in NDCmitiQ.

Advantages of the presented tool are, e.g., that it can be updated easily upon submission of new NDCs, and does not only provide estimates of regional / global emissions pathways but the national contributions and pathways. Furthermore, it can be run with different data sets of national emissions and socio-economic data. Currently, for simplicity estimates of the covered share of emissions are based on the main sectors, but as some NDCs name, e.g., only the sub-sector "Electricity Generation"
to be targeted and not the entire Energy sector, refinements could be implemented. Similar to Benveniste et al. (2018), targets for fossil fuel shares are not included in NDCmitiQ, and the non-fossil fuel targets the large emitters China and India stated additionally to emissions intensity targets are not quantified. Estimates of the international bunkers emissions and their planned mitigation are not addressed in NDCmitiQ. We restricted our uncertainty analysis on global scale to a limited set of options, generally changing one option at a time, to be able to trace back the changes to the single options. However, this analysis can
be further extended to address the interaction between the options, and quantify the resulting uncertainty range.

NDCmitiQ is limited in its capabilities to quantify NGT targets. For countries with this target type, the assumed mitigated emissions trajectory equals the baseline pathway. Only if the reclassified target type differs from NGT, the mitigated trajectory





in NDCmitiQ will differ from the reference emissions. In total, for 2017 / 2030 (dmSSP2) 5.5% / 6.1% of global emissions were emitted by countries classified as NGT (type_main). For type_reclass, the global shares are reduced to 3.5% and 3.8% for 2017

and 2030, respectively. Additional analyses and support for these NDCs would be beneficial for an improved quantification of the global mitigated emissions pathways. About 1% of 2017' emissions was emitted by countries without NDC, to which one must add the contribution by the USA (approximately 14%), who withdrew from the PA (all emissions excl. LULUCF and bunkers fuels). As for Parties with NGT targets, the baseline emissions are likewise assumed as mitigated trajectories for countries without NDC.

In the Paris Agreement it was decided that all countries should move towards economy-wide targets and raise their ambition over time. Based on the presented analyses, currently a total of 77% of global 2017 emissions are estimated to be covered by the NDCs (excluding LULUCF and bunkers fuels). As one of six countries, we assess that with the tested emissions and GDP scenarios, India's GHG mitigation target would show an unexpected behaviour when moving from the current estimated coverage towards a 100% coverage without simultaneously increasing the relative reduction level: it would result in a less

ambitious target, with noticeable impact on global scale.

Countries can use fixed baselines, which do not change over time and facilitate target and pathway quantifications (Graichen et al., 2018), but also leave room for over- or underestimation, as, contrary to dynamic baselines, the projected pathways are not adapted to parameter or methodology changes over the years. On global scale, for all historical years the baseline emissions data provided in the NDCs are lower than emissions from PRIMAP-hist, while for the year 2030 we find that they are +35 /

+98% (exclLU / inclLU) higher than the middle-of-the-road scenario dmSSP2. For a total of 97 countries (excl. USA) we were able to estimate targets based on NDC emissions data, and classify 77 NDCs as RBU targets (relative reduction against BAU emissions; target_orig), out of which 17 could not be quantified with NDC emissions data.

For the tested quantification options, the range of global mitigation pathways is dominated by the targets' conditionality and the underlying emissions and GDP data. Supporting findings by Benveniste et al. (2018) and Rogelj et al. (2017), we see a clear

influence of the assumed GDP, dominated by the fact that India and China pledged to reduce their emissions intensity per unit of GDP. In total, the analysed unconditional worst to conditional best emissions pathways differ by about 3.5–5.2 Gt $CO_2$eq in 2030 (10th / 90th percentiles for dmSSP1–4, median: 4.3 Gt $CO_2$eq). The effect of different quantification options, such as the covered share of emissions, or the evolution of emissions after the last target year (tested up to 2030), have a smaller impact on global scale. For the presented input data and quantification options, we estimate the global mitigated emissions in 2030 to

range between 49.2 and 55.7 Gt $CO_2$eq AR4 for dmSSP1–4 (10th / 90th percentiles, median: 52.4 Gt $CO_2$eq; excl. LULUCF and bunkers fuels).

## 5 Conclusions

Under the Paris Agreement, Parties agreed to limit global warming to 1.5–2°C, but studies show that the current set of NDCs does not put us on track to reach this temperature goal. The quantification of GHG mitigation targets is ongoing research, as

Parties are expected to regularly submit updated NDCs. The new open-source tool NDCmitiQ can be used for target quantifi-





cations, and to derive national and global emissions pathways consistent with the NDCs. The emissions pathways can serve as basis to estimate, i.a., the 2100 temperature rise. To get a better picture of the range of possible outcomes from a full implementation of the NDCs, it is of advantage that various institutions quantify the mitigated emissions, as they include uncertainties and often result in an estimated emissions range. Examples for uncertainties are addressed in NDCmitiQ and presented in the manuscript, such as: the conditionality of targets; the underlying emissions estimates and socio-economic data; the share of national emissions covered by an NDC; or uncertainties from LULUCF. More clarity in the NDCs on the described issues would narrow down the range of quantified national and global mitigated emissions, here estimated to range between 49.2–55.7 Gt $CO_2$eq AR4 in 2030 for SSP1–4 ($10^{th}$ / $90^{th}$ percentiles, median: 52.4 Gt $CO_2$eq; excl. LULUCF and bunkers fuels).

*Code and data availability.*

We use a GitHub repository to work on the Python-based tool to quantify GHG mitigation targets and emissions pathways NDCmitiQ (`https://github.com/AnnGuenther/ndc_quantifications.git`). All data sets we produced with NDCmitiQ for the presented manuscript, and the code version NDCmitiQ v1.0.0 are available for download on zenodo (Günther et al., 2020). For each quantification (about 1min20s run time) one folder is provided, with the folder name structure being `ndcs_yyyymmdd_hhss_`, followed by:

**SSP1 to SSP5:** which SSP marker scenario is chosen for the run. This information is also important if the run is based on NDC emissions data (type_reclass), as not for all countries emissions data were provided, and the SSP baselines are used for the pathway construction.

**typeReclass:** runs with type_reclass, based on emissions data from the NDCs where possible.

**typeMain:** runs with type_main, based on external emissions data (PRIMAP-hist v2.1 HISTCR and down-scaled SSP marker scenarios).

**pccov100:** runs with an assumed coverage of 100%.

*Without pccov100*: coverage based on estimated %cov (Sect. 2.3).

**constEmiAfterLastTar:** runs with assumed constant emissions after a Party's last target year.

*Without constEmiAfterLastTar*: instead of the emissions, the relative difference to the baseline is kept constant after the last target year.

**BLForUCAboveBL:** runs using the baseline emissions as the unconditional pathways for Parties without unconditional targets, even if the baseline is better than the conditional targets.

*Without BLForUCAboveBL*: conditional worst pathway is used in this case instead of the baseline.

**UNFCCC / FAO:** runs using LULUCF data with UNFCCC or FAO chosen as the primary prioritised data source (UNFCCC, CRF, BUR, FAO or FAO, CRF, BUR, UNCFFF).

*Without UNFCCC / FAO*: prioritisation is CRF, BUR, UNFCCC, and FAO.

Per run, the single per-country targets can be found in `ndc_targets.csv`, the country-pathways are available in `ndc_targets_pathways_per_country_used_for_group_pathways.csv`, and the aggregated pathways are stored in `ndc_targets_pathways_per_group.csv`. Additionally, each of the folders contains the file `log_file.m` (information on the setup for the model run), and the sub-folder `/per_country_info_on_target_calculations/` that provides per-country information on how exactly the national targets were quantified. The time series that serve as input to the quantifications can be found in the folder `/data/preprocess/`, together



with the estimated coverages. Information on how to use NDCmitiQ is provided in the repository (`README.md`, `requirements.txt`,
`/docs/build/html/index.html`). The input that can easily be modified is: time series of emissions (exclLU and onlyLU), %cov
(exclLU), population, and GDP, and information from the NDCs.

## Appendix A:  Additional information

In the Appendix, additional and explanatory information is given as referenced in the manuscript.

### A1    Pre-processing of projected non-LULUCF emissions

Pre-processing of the five down-scaled SSP marker scenarios (dmSSP1–5) is performed to fill missing time series for some
countries (information for Sect. 2.2.1). For a few countries, data are not available for all five dmSSPs (emissions, population
and GDP: up to six countries, with a global share of up to 0.1% in 2017), in which cases the missing dmSSP is approximated
as the mean time series of all available dmSSPs.

Up to 43 countries (depending on the scenario and entity) with very small emissions, population, or GDP, have no down-
scaled time series available for dmSSP1–5, representing global shares of merely up to 0.2%, 0.5%, and 0.1% in 2017, respec-
tively. For these countries, estimates for future years are based on linear regression to the PRIMAP-hist data in 2012–2017.

Some countries only cover certain F-gases in their mitigation targets, and depending on the target type we might need
scenarios of the single contributions for the calculation of the covered share of emissions. As for dmSSPs, no information is
available on the contribution of HFCs, PFCs, $SF_6$ and $NF_3$ to the total basket of F-gases, we base our estimates on the historical
contributions (mean over 2012–2017).

### A2    Emissions time series for LULUCF (non-NDC data)

The LULUCF data sources included in NDCmitiQ are prioritised as follows:

**CRF 2019 and CRF 2018**  Emissions data reported to the UNFCCC by former Annex-I countries (industrialised countries)
in the **C**ommon **R**eporting **F**ormat (UNFCCC, 2019c, 2018; Jeffery et al., 2018a, b; Gütschow et al., 2020). The year
indicates the submission year.

**BUR 3, BUR 2 and BUR 1**  Emissions data reported to the UNFCCC by former non-Annex-I countries in their **B**iennial
**U**pdate **R**eports (UNFCCC, 2019b). BUR 1 are the first and BUR 3 are the most recent submissions (if available).

**UNFCCC 2019**  National Communications and National Inventory Reports for developing countries (UNFCCC, 2019a).

**FAO 2019**  FAOSTAT database: Food and Agriculture Organization of the United Nations (FAO, 2019).

For some countries, only FAO or UNFCCC or CRF have LULUCF emissions data available, for other countries FAO &
UNFCCC, or FAO & CRF, or FAO & BUR have data, while for others FAO & UNFCCC & BUR provide data. A country's
LULUCF emissions time series from the chosen data source are interpolated linearly and then extrapolated constantly, by the





mean over 1990–1997 if backward extrapolation is necessary, and the mean over all data points starting in 2010 as projected
LULUCF emissions. Regarding forward extrapolation, the approach is similar to one of the LULUCF scenarios in Fyson and

795 Jeffery (2019), with the average over 2004–2014 in their case. Our LULUCF data time series do not take into account current
or planned afforestation, deforestation, or reforestation plans. For some Parties, country-reported data have no values available
between 1990–1997 or after 2009, and in these cases, extrapolation is based on the first / last available value only. Due to
extremely scarce country-reported data availability in some cases, we chose to merely dismiss time series from a certain data
source for a country if less than three data points are available for 1990–2017. However, if none of the other sources has at least

800 three annual values available, the source with higher prioritisation is used nonetheless. This check does not consider whether
available values differ on an inter-annual basis, so time series are not dismissed if they were interpolated beforehand. We use
available Kyoto GHG emissions or the sum over the relevant gases in the LULUCF sector, $CO_2$, $CH_4$, and $N_2O$ (inter- and
extrapolation prior to aggregation).

CRF 2019 is chosen for 43 countries (-1 822 Mt $CO_2$eq; for 2017 and GWP AR4), BUR 3 for two countries (-102 Mt $CO_2$eq),

BUR 2 for seven countries (811 Mt $CO_2$eq), BUR 1 for three countries (-7 Mt $CO_2$eq), UNFCCC 2019 for 47 countries (-
2 975 Mt $CO_2$eq), and FAO 2019 for 93 countries (2 000 Mt $CO_2$eq). As all countries with CRF 2018 data already submitted
their CRF 2019 data, CRF 2018 is not actually used. However, for future updates, it makes sense to include the option to chose
data from the previous year, in case not all countries have yet submitted new data.

## A3 Baseline emissions time series based on NDC-data

We collected the emissions baseline data from within NDCs and classified them as excluding LULUCF, including LULUCF,
and only LULUCF (exclLU, inclLU, onlyLU; details to Sect. 2.2.3). Additionally, based on these emissions and complemented
by the PRIMAP-hist and dmSSP emissions, a data set of national emissions time series (exclLU) was constructed. To start with,
the details on how we choose which LULUCF emissions to use for the target quantifications are given in Table A1.

**Table A1.** Decision making on which emissions data to use for LULUCF. The following is checked in the presented order, and the first match
is used as $onlyLU$ emissions.

| | | |
|---|---|---|
| If | onlyLU emissions are provided in the NDC | $onlyLU = onlyLU_{NDC}$ |
| Else, if | inclLU and exclLU data are provided in the NDC | $onlyLU = inclLU_{NDC} - exclLU_{NDC}$ |
| Else, if | inclLU data are provided in the NDC | $onlyLU = inclLU_{NDC} - exclLU_{external}$ |
| If any of the above is true, and onlyLU emissions data are available for the period 2010–2017 but not for years | | |
| after 2017, use constant extrapolation to future years based on the average over the available values in 2010–2017 | | |
| Else | use the external LULUCF emissions data | $onlyLU = onlyLU_{external}$ |

To create mitigated emissions pathways, we intend to use target emissions that exclude contributions from LULUCF, and

815 construct a data set spanning 1990–2050 based on the data provided within NDCs, combined with the PRIMAP-hist and SSP
emissions for completeness, that can then be used for pathway creation. The constructed data set is based on the PRIMAP-hist
v2.1 HISTCR Kyoto GHG national emissions time series (exclLU) up to 2017, followed by the $exclLU_{NDC}$ emissions, with





linear interpolation between 2017 and the available NDC data. If the last year of $\mathrm{exclLU_{NDC}}$ is earlier than 2050, we use the dmSSP growth rates to extrapolate the emissions pathway, resulting in one constructed data set per dmSSP (further details in Table A2). Even though, up to 2017, the NDC data set is constructed with PRIMAP-hist emissions, the emissions given within NDCs are used to quantify their targets (for type_reclass), unless it is stated otherwise, e.g., for comparison runs (type_main).

**Table A2.** Details on the approach used to construct an "NDC emissions data set (exclLU)" for 1990–2050.

| Up to 2017 | PRIMAP-hist v2.1 HISTCR Kyoto GHG national emissions (exclLU) | $\mathrm{exclLU} = \mathrm{exclLU_{external}}$ |
|---|---|---|
| After 2017 | If NDC provides emissions exclLU | $\mathrm{exclLU} = \mathrm{exclLU_{NDC}}$ |
| | Else, if NDC provides emissions inclLU (onlyLU estimated as described above) | $\mathrm{exclLU} = \mathrm{inclLU_{NDC}} - \mathrm{onlyLU}$ |
| | Fill gaps by linear interpolation and if necessary extrapolate the pathway using the growth rates from the current down-scaled SSP marker scenario. | |

### A3.1 Covered share of emissions

The quantification rules for the share of emissions covered by a Party's NDC GHG mitigation target (%cov, excl. LULUCF) are given Table A3 (details for Sect. 2.3). In general, %cov is based on an assessment of the covered main sectors and GHGs, and on PRIMAP-hist emissions data per sector and gas combination (years up to 2017). For the period after 2017, estimates are the average recent %cov, or derived from the correlation between covered and total national emissions (all for 2010–2017). The applied rules are further clarified in Table A4, and Figure A1 contains per-country information of the covered sectors and gases. The coverage is presented as provided (more or less explicitly) in the NDCs, and as "adapted" for the use in NDCmitiQ. Results for %cov were used in Geiges et al. (2019), with small changes in the methodology since then.

Estimates of %cov for upcoming years, needed to define the (not-)covered emissions share in the target years, are based on the decisions and quantifications outlined in Figure A2. Either the average recent values of %cov are kept constant or estimates are calculated from the correlation between national total emissions and %cov (2010–2017). NDCmitiQ provides two options as projection preference: "correlation" (default) or "mean". The scheme presented in Figure A2 describes the steps if mean is chosen as preference. For the option "correlation" the correlation is used for each country, unless the r-value of the regression line to the correlation is below a defined limit (0.85). If the correlation is used, the estimates of %cov depend on the projected national emissions and therefore on the chosen dmSSP scenario.

In Table A5, the national shares of emissions per sector / gas are given as 95[th] percentiles, to reduce the influence of extreme values and missing data. Further, the number of countries assessed to cover emissions from a certain sector / gas are provided. The information is complementary to Table 6.

### A4 Options for the calculation of emissions pathways

Several options to modify the calculation of emissions pathways are implemented in the tool.

**Targets only for countries X, Y, Z:** Use quantified targets for countries X, Y, and Z, else use baseline emissions.





**Table A3.** How we define the share of emissions covered by an NDC (%cov; excluding LULUCF). "economy-wide" stated: all sectors (LULUCF treated separately) are assumed to be covered, even if a list of covered sectors is given that is not complete. If in the NDC it becomes obvious, however, that the reduction merely applies to emissions from certain sectors, only these sectors are covered. Example on decision making from box 1+2 in Table A4 (Sect. A3.1).

---

**Coverage & %cov**
**National part of covered emissions**
**(excluding LULUCF)**

**Single sectors and gases:**
- The stated sectors and gases are set to covered, the remaining sectors and gases are assumed not to be covered.
- If *nothing* is stated on covered *gases*: $CO_2 + CH_4 + N_2O$ assumed to be covered.
- If some *F-gases* are covered, *IPPU* is assumed to be covered, even if not stated so.
- If all sectors are covered we assume emissions from '*Other*' to be covered.
- If *economy-wide* is stated, all sectors exclLU are assumed to be covered,
  unless it is clear that the reduction only applies to certain sectors.

**Emissions from a sector plus gas combination:**
- If the sector or gas is classified as not-covered: these emissions are counted as not-covered.
- Else, the emissions from this sector plus gas combination are counted as covered.

**Quantification** of *%cov* needed for the historical base years (for RBY and REI_RBY) and the target years.
*Historical years:* based on single gas plus sector combinations and PRIMAP-hist v2.1
  HISTCR emissions data (available per gas plus sector combination, up to 2017).
*Future years:*
  - *All gases & sectors covered:* 100%.
  - *Not all gases covered:*
    - *All sectors covered:* based on national SSP pathways per gas.
      Available for CO2, CH4, N2O and F-gases. F-gases are split by PRIMAP-hist mean shares*.
    - *Not all sectors covered:* PRIMAP-hist averages* kept constant or estimates based on the correlation between
      historical national emissions and covered part of emissions*.

\* for the period 2010 - 2017

---

**Table A4.** Decisions on covered sectors and gases. "+" = "covered", "–" = "not-covered", and "/" = "no information available".

| Gas / sector | | | Energy | IPPU | Agriculture | Waste | Other |
|---|---|---|---|---|---|---|---|
| | Given in NDC | | + | / | + | / | / |
| | "Adapted" | | + | + (as $SF_6$ is "+") | + | − | − (as not all "+") |
| $CO_2$ | + | + | + & + = + | + & + = + | + & + = + | − & + = − | − & + = − |
| $CH_4$ | + | + | + & + = + | + & + = + | + & + = + | − & + = − | − & + = − |
| $N_2O$ | + | + | + & + = + | + & + = + | + & + = + | − & + = − | − & + = − |
| HFCs | / | − | − & + = − | − & + = − | − & + = − | − & − = − | − & − = − |
| PFCs | / | − | − & + = − | − & + = − | − & + = − | − & − = − | − & − = − |
| $SF_6$ | + | + | + & + = + | + & + = + | + & + = + | − & + = − | − & + = − |
| $NF_3$ | / | − | − & + = − | − & + = − | − & + = − | − & − = − | − & − = − |





**Figure A1.** Sectors and Kyoto GHGs covered by NDCs on a per-country level. Crosses: +/- explicitly mentioned coverage, squares: adapted coverage used in NDCmitiQ. EU target information: shown for single countries (e.g., Germany). The per-country share of global Kyoto GHG emissions is presented (for 2017, based on PRIMAP-hist v2.1 HISTCR, GWP AR4, excl. LULUCF, and bunkers fuels). Shares displayed in green: target is intended to be economy-wide. USA: only shown for information purposes.



**Future estimates of the part of national emissions covered by an NDC (non-LULUCF emissions)**

| All sectors & all gases covered | 100% | All sectors & not all gases covered | Sum over covered gases, using SSP data per gas. F-gases split by historical average shares*. |
|---|---|---|---|

**Not all sectors covered & (not) all gases covered**

Mean over recent years* kept constant
or
pc_cov calculated based on correlation between total and covered° emissions*

If mean over historical pc_cov* > 90%: keep the mean constant.
If mean over historical pc_cov* < 10%: keep the mean constant.

else:

Slope of linear regression to historical values of pc_cov*

abs(slope) < threshold (little variability): mean* kept constant

abs(slope) > threshold (higher variability): testing the correlation between total and covered° emissions*

r-value of linear regression > threshold & pc_cov 2018 - 2050 between 0% and 100%: use correlation

else:

use mean over recent years*

*Correlation*:
- Future pc_cov derived from the linear regression to the covered° part of emissions vs. the total emissions for 2010 - 2017, and the estimates of future total annual emissions and the corresponding covered° emissions.
- No too drastic changes expected in pc_cov over time.
 - If future values of pc_cov > 90% but all of them < 90% in 2010 - 2017: values < 90% set to 90%.
 - If future values of pc_cov < 10% but all of them > 10% in 2010 - 2017: values > 10% set to 10%.

Threshold for *slope*: +/- 1%; limit for *r-values*: 0.85 (both based on testing).
° if mean historical pc_cov* < 50% the correlation between the total and not-covered part of emissions is used.

* for 2010 - 2017.

**Figure A2.** Projections of the covered share of national emissions: scheme on decision making if the preference for the calculation of future values of %cov is set to "mean". Else the correlation is used for all countries unless the r-value of the regression line to the correlation is below 0.85.

**Prioritised target types:** Use prioritised target types for countries X, Y, and Z with the target types being in the order A, B, and C, else use type_reclass. type_main: use the 'main target type' (what has been stated +/- in the NDC as target type); type_reclass: use the reclassified target type (mostly ABS with the quantification based on data from the NDCs).

**Countries without unconditional targets & what if the baseline is better than the conditional targets:** Use the baseline emissions as unconditional pathways if no unconditional targets are available and the conditional worst pathway in 2030 is worse than the baseline emissions in 2030. Default: instead of the baseline, the conditional worst pathway is used as unconditional pathways as well.

**Set coverage (exclLU) to 100%:** For a set of countries or all countries.

**Strengthen targets:** Strengthen targets by a certain percentage P, for countries X, Y, and Z. Either by adding P to the value given in a country's NDC, or by multiplying the reduction by $100\% + P$. If the resulting percentage is greater than 100%, it is set to 100%, which means a total reduction of the – covered share of – emissions. For example, for a target with 20% reduction and P $= 10\%$, if "add" is chosen the result is $- (20\% + 10\%) = -30\%$, and if "multiply" is chosen the result is $-20\% \cdot \frac{100\% + 10\%}{100\%} = -22\%$. For absolute targets (ABS, AEI, ABU), it is not distinguished between add and multiply. In the case of ABS and AEI, the strengthened target is, e.g., $20 \text{ Mt CO}_2\text{eq} \cdot \frac{100\% - 10\%}{100\%} = 18 \text{ Mt CO}_2\text{eq}$, or





**Table A5.** Relative contribution of different gases and sectors to the national 2017 Kyoto GHG emissions (95[th] percentiles, part a), and number of countries for which a gas / sector is covered by its NDC (part b). **(a)** 95[th] percentiles for the national shares of emissions from a certain gas / sector (e.g., Energy-emissions: in 95% / 5% of the nations, Energy-emissions represent less / more than 91.5% of national emissions). All values exclude emissions from LULUCF and bunkers fuels emissions. All values are based on PRIMAP-hist v2.1 HISTCR emissions data (GWP AR4). **(b)** Coverage from within NDCs (more or less explicitly stated) and "adapted" coverage (based on the rules described in Sect. 2.3). E.g., IPPU / $CO_2$: 123 / 174 countries are assessed to +/- clearly state in their NDCs that they cover their IPPU / $CO_2$ emissions, with the adapted number of countries being 142 / 193. This results in 142 countries assessed to cover their $CO_2$ IPPU emissions. USA: counted as 'no coverage'; EU: countries counted as single countries. Countries with NGT targets that state covered sectors and gases are included in the presented numbers.

**(a) Relative contributions to national emissions (95[th] percentiles), per gas & sector**

| 2017 | | $CO_2$ | $CH_4$ | $N_2O$ | HFCs | PFCs | $SF_6$ | $NF_3$ |
|---|---|---|---|---|---|---|---|---|
| | | 91.5% | 65.8% | 28.6% | 8.9% | 1.1% | 0.5% | 0.0% |
| **Energy** | 91.5% | 87.2% | 29.3% | 1.5% | – | – | – | – |
| **IPPU** | 17.5% | 13.8% | 0.2% | 0.9% | 8.9% | 1.1% | 0.5% | 0.0% |
| **Agriculture** | 70.8% | 1.0% | 47.3% | 27.4% | – | – | – | – |
| **Waste** | 20.3% | 0.4% | 19.4% | 1.1% | – | – | – | – |
| **Other** | 1.3% | 0.0% | 0.0% | 1.3% | – | – | – | – |

**(b) Gases and sectors covered by NDCs**

| 2017 | *NDCs (Adapt.)* | $CO_2$ | $CH_4$ | $N_2O$ | HFCs | PFCs | $SF_6$ | $NF_3$ |
|---|---|---|---|---|---|---|---|---|
| *NDCs (Adapt.)* | | *174 (193)* | *157 (175)* | *147 (165)* | *78 (79)* | *71 (72)* | *71 (73)* | *51 (53)* |
| **Energy** | *193 (193)* | 193 | 175 | 165 | – | – | – | – |
| **IPPU** | *123 (142)* | 142 | 135 | 132 | 79 | 72 | 73 | 53 |
| **Agriculture** | *139 (143)* | 143 | 137 | 135 | – | – | – | – |
| **Waste** | *149 (148)* | 148 | 142 | 135 | – | – | – | – |
| **Other** | *0 (124)* | 124 | 118 | 117 | – | – | – | – |

$10 \text{ t CO}_2\text{eq} \cdot \frac{100\% - 10\%}{100\%} = 9 \text{ t CO}_2\text{eq}$, while in the case of ABU the calculation follows, e.g.,

$-2 \text{ Mt CO}_2\text{eq} \cdot \frac{100\% + 10\%}{100\%} = -2.2 \text{ Mt CO}_2\text{eq}.$

*Author contributions.* AG designed the study, implemented the module, carried out the analyses, and led the manuscript writing process. All
860 authors discussed the methodology and results and contributed to the presented manuscript.

*Competing interests.* We have no competing interests to declare.



*Acknowledgements.* AG and JG acknowledge support by the German Federal Ministry for the Environment, Nature Conservation and Nuclear Safety (16_II_148_GLobal_A_IMPACT). AG and JG further acknowledge support by the German Federal Ministry of Education and Research (01LS1711A).





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
