# Peer review of "NDCmitiQ v1.0.0: a tool to quantify and analyse GHG mitigation targets"

_Geoscientific Model Development, 2020_

## Author Response (AR1)

Dear Referees,

We wish to repeat our thanks that you reviewed this long and detailed paper, and for your feedback that helped us to improve the model and the manuscript. While, as this is a model description paper, the methods section remains an integral focus of the manuscript, we worked on a revised version considering where the paper can be shortened and rearranged in a way that helps balancing the "methods" and "results / discussion" sections. Even though still resulting in a rather technical paper, we believe that showing how many decisions and steps are needed to derive global emissions pathways based on the NDCs' mitigation targets increases the understanding of uncertainties connected to such emissions pathways, and indicate why a clear communication of targets is necessary to reduce such uncertainties. In line with your indications we expanded the discussion of model shortcomings, and additionally the advantages of our approach. This tool is meant for all countries and with this broad scope of countries we do not perform detailed policy analyses on country-level. In the manuscript we acknowledge this fact more explicitly now.

Please find more details on our changes in response to your remarks below (referee comments in black, our responses in blue), as well as the revised manuscript with all changes being marked (blue: added text; red: removed text; additionally, we use cyan for comments on moved Tables and Figures, as with track changes, all moved Tables and Figures plus their captions would have been coloured, making it difficult to identify changes within the captions or Tables).

Best regards, Annika Günther (OBO my co-authors)

**General information:**

For a more balanced manuscript, we removed several Tables and Figures from the main text and added them in the Annex. Namely former Table 2, Table 3 plus parts of the text in this section, Table 6.a and Figure 5a-d plus parts of the text in this section. Furthermore, Table 7 was shortened while the complete Table was moved to the Annex.

In general, our manuscript is still based on the NDCs submitted by mid-April 2020, in line with the discussion paper, but we included our updated assessments of NDCs submitted by end-December 2020 in Section 3.4 (see changes in Sect. 2 + 3.4).

To better specify which NDC submissions should be considered while running NDCmitiQ, we included the option "submissions_until" to indicate submissions up to which date to consider. Per country, the newest submission up to the given date is used. In such a way, it is easier to track progress on national to global scale, by performing two runs with the same settings, in which only the submissions_until date is changed (see Sect. 3.5).

We included the option "constant_difference" for the pathway creation, meaning that the baseline growth rate is used and the difference in absolute emissions of the country's mitigated pathways to the chosen baseline trajectory after the country's last target year is kept constant (see changes in Section 2.4.2 and 3.4).

As we do not perform policy analyses and for comparison purposes, we further included the option to choose targets as provided by the Climate Action Tracker (www.climateactiontracker.org) for specified or all countries with estimates available from the CAT when constructing the globally aggregated mitigation pathways. Therefore, we extracted all NDC target estimates, and apply the pathway construction methods presented in this manuscript (Sect. 2.4.2) to derive national mitigated pathways from the given point values. The respective information was added in Section 2.4.2.

**CEC1**

Dear authors,

In the next step of the review process, please be sure that you include in the Code availability section the DOI and link to the Zenodo web page containing the code of the model.

Also, the statement on your use of GitHub is irrelevant and can lead to confusion. Please remove this sentence 'We use a GitHub repository to work on the Python-based tool to quantify GHG mitigation targets and emissions pathways NDCmitiQ (https://github.com/AnnGuenther/ndc_quantifications.git).'

Many thanks,

Juan A. Añel

Geosc. Mod. Dev. Executive Editor

Dear Editor, thank you for these comments, we changed the manuscript accordingly.
* * *
**RC1**

Dear Referee 1 (RC1), we very much appreciate your detailed comments regarding the NDCmitiQ methodologies and also its shortcomings. Your comments helped us to perform changes in the tool, and to extend the paper's discussion section. Please find more details below.

This is an interesting analysis on an uncertainty analysis for the impact of the uncertainties related to the INDCs on the global emission levels. It includes many detailed analysis and insights, which are well described. The work is highly relevant and interesting, and also the tool looks promising. The paper itself is rather detailed and lengthy, and in my view reads more as a technical report, than a journal paper.

However, I see some short-comings, which in my view can highly influence the resulting global emissions projections. Unfortunately the results of some main emitting countries, such as China and India are not included in the paper, so I could not check the projections.

In general the NDC emissions projections differ across studies mainly due to a couple of important factors.

For a more balanced version of the paper, we moved several Tables and Figure and parts of the text to the Annex. Nevertheless, as this is a model description paper and we want to show how difficult it is to quantify the mitigation targets and how many assumptions can be needed during the process, we decided against removing large parts of the content from the main text.

Our estimates for India are included in the manuscript (Section 3.3), though not for China, for which results can be accessed at https://doi.org/10.5281/zenodo.4744655 (together with all remaining quantification results).

1/ The authors assumed in their calculations that the NDC targets of China and India is calculated in terms of carbon intensity improvement. They mention: "Similar to Benveniste et al. (2018), targets for fossil fuel shares are not included in NDCmitiQ, and the non-fossil fuel targets the large emitters China

and India stated additionally to emissions intensity targets are not quantified.". The NDC of China also includes (i) the target to peak $CO_2$ emissions by 2030 at the latest, (ii) increase the share of non-fossil energy carriers of the total primary energy supply to around 20% by that time, and (iii) increase its forest stock volume by 4.5 billion cubic metrics, compared to 2005 levels. In fact the factor (i) and (ii) are more important for the final 2030 emissions than the intensity target, as factor (i) and (ii) are the dominant factor. See literature around this issue from climate action tracker, but also UNEP Gap rapport, etc.. This also holds for India, since the NDC target of India also includes (i) to increase the share of non-fossil based power generation capacity to 40% of installed electric power capacity by 2030 (equivalent to 26–30% of generation in 2030), and (ii) to create an additional (cumulative) carbon sink of 2.5-3 GtCO2e through additional forest and tree cover by 2030.

For the calculation of the impact of the NDC for China and India the authors need to account for the factors (i) and (ii) for China and (i) for India, and this highly affects the outcomes, as these factors are more dominant than changes in the GDP. Accounting for these interactions in the calculations would significantly change the result of the analysis, and the impact of the uncertainties in the GDP projections would be much less.

This is not easy, as you need to account for energy model calculations, and since the authors have used different model projections for the SSP scenarios, and the authors may not have access to all energy calculations, I foresee a difficult issue here how to improve the analysis. However, I think this issue needs to be addressed, as the current analysis overestimates the impact of uncertainties on the projections, and it leads to rather high NDC emissions projections.

The authors refer to Benveniste et al., but this study is an outlier in the range of NDC studies, mainly due to the high emissions projection of China. Benveniste et al also do not include current policies and all NDC targets.

Unfortunately I could not find any details on the NDC emissions projection of China, so I could not check this.

You raise very interesting points, which are indeed very challenging. Our tool has the limits you described and does not account for energy model calculations. Such detailed assessments are outside the scope of NDCmitiQ. The model is intended to perform a broad analysis across multiple countries and other institutes are more focussed on tailoring the approach to individual countries. Each approach has its strength and weaknesses, and NDCmitiQ adds capabilities to the suite of analyses available. In order for us to include energy model calculations, we would need to have data available on country-level, and matching to the chosen baselines (SSPs), which are not available in the SSP database. For the emissions factors we would only have default values from the IPCC reporting guidelines but more importantly there is a lack of available energy mix data for the SSPs. It might be feasible for us to derive rough estimates of countries' energy targets. Even when considering the CMIP6 data with more regions and higher sectoral detail available, with which estimates might therefore be feasible, we do not currently plan to extend NDCmitiQ in such a way. We included a statement regarding these issues in the discussion section (Sect. 4). To provide these rather rough estimates which would nevertheless imply a lot of additional work is outside the scope of our tool.

Following your comments, we added the option to use pre-calculated NDC assessments for individual or all countries (for which data are available) based on the targets provided by the Climate Action Tracker, who do consider China's and India's energy-related targets.

As noted above, for China our quantification results can be accessed at https://doi.org/10.5281/zenodo.4744655. Additionally, as mentioned in the manuscript, we do

consider if a country indicated an emissions peak (see Sect. 2.1 and 2.4.2), and added this information regarding China's target more explicitly now.

2/ I would also recommend that the authors use as a starting point a current policies scenario, and not the SSP no policy scenario. Some SSP scenarios do not include impact of current policies that are adopted after 2005 or 2010, and these scenarios are rather hypothetical scenarios, and not very realistic. As mentioned above, you can better account for the current policies in the NDC calculations for India and China, but also for many other countries.

Some SSP scenario lead to very high short-term emissions, which are highly criticized in the literature, see: Hausfather, Zeke, and Glen P. Peters. "RCP8. 5 is a problematic scenario for near-term emissions." Proceedings of the National Academy of Sciences 117.45 (2020): 27791-27792.

The intention behind NDCmitiQ is not to quantify what is happening / will actually happen, for which we would need to perform policy based assessments and better consider current policy scenarios instead of the SSP baseline projections, but to estimate what would happen based on what is said in the NDCs and provide a plausible emissions range. As this often results in uncertain estimates and emissions ranges, we consider the SSP baseline projections as adequate options, which do include different levels of policies. In a future setup, we will consider to include available scenarios that consider more recent policies to assess their implications on the quantification results. As NDCs are often stated against counter factual no-policy baselines the use of current policy scenarios can overestimate the mitigation for these NDCs.

RCP8.5 is at the upper edge of high emissions scenarios without mitigation measures, and has been criticised in recent years not to represent near-term emissions, and for its use as a baseline projection in comparison to lower RCPs used as mitigation scenarios. We here use the SSP5 marker scenario as one baseline option for target quantifications. As pointed out by Hausfather and Peters (2020), "The SSPs – which are being used by researchers going forward – show that the SSP4-6.0 and SSP2-4.5 scenarios agree much better with near-term cumulative emissions than the SSP5-8.5 scenario". For the median estimates of global mitigated emissions presented in Section 3.4, we only included SSP1-4, which is in line with the critique for RCP8.5 and SSP5 which comes closer to RCP8.5 than SSP1-4. In Section 3.4, we included a clarifying statement.

3/ How does this study include surplus emissions? The global NDC emissions projections from various NDC studies excludes the impact of surpluses, so if the current policies projection for a country is below the NDC target, the NDC emissions projection is equal to the current policies scenario. This has a large impact on the global emissions projections, in the order of 2-3 GtCO2e. This issue is not only relevant for India, China, Turkey, which overachieve their NDC target, but also for some countries with lower emissions projections. For me, it is not fully clear how the authors includes this impact, and in my view, it would lower the global NDC emissions projections, in particular if the analysis uses as a starting point current policies scenarios.

In the setup presented in the discussion paper, we did not really consider this fact. The given or calculated target emissions are used for a country, even if the target lies above the assumed baseline for un-/conditional cases. However, as it is indeed a very interesting point, we updated the tool with the option to use the baseline emissions in case the target lies above the baseline. Comparing the default setting with such a quantification enables us to roughly estimate surplus emissions (Sect. 2.4.2 + 3.4).

4/ Land use emissions. I agree that it is very challenging to include LULUCF in the projections, but it is an important source of uncertainty that needs some discussions. I noticed the -2.1 GtCO2 estimate, which seems rather low compared to the analysis of the LULUCF inventory data by Grassi et al. (2017;

2018) in Nature Climate Change, which discusses the impact of the LULUCF data in the NDC emissions projections in much detail. Can you explain why your estimate falls outside the range presented by Grassi et al..

We agree that LULUCF is a very difficult aspect of NDCs and mitigation targets. LULUCF emissions and removals are difficult to measure, report and check. Grassi et al. (2017) estimated the land use and forests' net anthropogenic source for 1990-2010 to be 1.3+/-1.1 GtCO2eq/yr (using historical data from UNFCCC ((I)NDCs, 2015 GHG inventories, National Communications); other official countries' documents; and finally FAO-based data sets), which is indeed higher than our estimates - unless prioritising FAO data.

We updated the LULUCF table to a Figure showing the ranges from 1990-1999, 2000-2009, 2010-2017, and the projected estimates, rather than annual values for 1990, 2010, 2017, and 2030. As we are looking at a global scale but not all data sources include data for all countries, we chose a prioritisation order of data sources to choose the country-level data from. In our setting, the default prioritisation order is CRF2019, CRF2018, BUR3IPCC2006I, BUR2IPCC2006I, BUR1IPCC2006I, UNFCCC2019BI, and FAO2019BI. To get a better understanding of the range of available values, we assessed all possible combinations of these data sources as source prioritisation orders, and the results are included in the updated manuscript. Here, one has to keep in mind that this results in a certain bias, as for example FAO, which is at the upper edge of estimates, only appears as data source once and therefore has less weight in the data source permutations, while the country-reported data are at the lower edge and have a higher weight in this assessment.

We conducted further analyses to understand the differences between our and Grassis's estimates. Even though we could not find a clear methodological explanation for the discrepancies, we see changes in total sums for different data versions of a data source. For CRF; UNFCCC, and FAO, we compared data versions from different years (see Fig. 1 below), with linear inter- and extrapolation as described in the manuscript applied before aggregating the available country-level data. We only sum data from countries which are available in each version of a data source, respectively. Not all data sources cover the same period of time, and the extrapolation is based on the mean over 2010-2017 or the last available value if none are available in 2010-2017. While for CRF the newest considered version (CRF2019) mostly has the lowest values (strongest sink), for FAO the newest considered version (FAO2019) mostly has the highest values (strongest source), compared to the previous versions, respectively. Therefore, the fact that Grassi et al. (2017, 2018) use older data versions than we use in our manuscript can explain some discrepancies in the global emissions estimates.

[Figure]

Figure 1: Kyoto GHG LULUCF emissions from different data sources and versions. Per data source (CRF, UNFCCC, FAO), country-level data aggregates from all countries for which data are provided in all versions, respectively (noted above each panel). The given categorisation name "IPCMLULUCF"

follows the IPCC 2006 categories and Jeffery et al. (2018, https://doi.org/10.5194/essd-10-1427-2018), while "CAT5" follows the IPCC 1996 categorisation.

Detailed comments:

Line 246: JRC? Why do you refer to JRC?

We changed the citation to IPCC AR5 WG3 Chapter 11, where these difficulties are stated as well.

**RC2**

Dear Referee 2 (RC2), we want to thank you very much for taking the time to give us feedback on the presented manuscript. Below, we address your comments.

The manuscript presents in a nice and comprehensive way substantial amount of work the authors completed to develop the tool, which includes several layers and steps to ensure the quality of the tool and the results. This tool is of high scientific relevance, considering the need to monitor and quantify progression overtime of global ambitions in climate change mitigation efforts, and given its open-access nature, it could become a very important modelling resource for researchers dealing with global emissions modelling. However, in its current form, the manuscripts focuses disproportionately on the methods and could be improved with a more balanced version that increases the weight and importance of the results and discussion sections.

For a more balanced version of the paper we moved several Tables and Figure and parts of the text to the Annex (details to be found above and in the marked down revised manuscript). We submitted the manuscript to GMD as we think that the journal gives us the opportunity for such a detailed model description, in combination with some results and discussions. We intended to show how difficult it is to quantify the mitigation targets and how many assumptions can be needed during the process, and therefore decided against removing large parts of the content from the main text.

Regarding the aim, relevance and reach of the manuscript and tool, I think the authors should limit more explicitly the audience to research and modelling community and be more concrete about the potential applications of the tool, which mostly relate to comparing on an equal basis mitigation pledges overtime and monitoring progress towards global mitigation ambition. Also the shortcomings of the tool, in particular for interested stakeholders should be mentioned more explicitly. For instance, while many stakeholders interested in tracking and monitoring mitigation ambition can be interested in keeping specific countries accountable on their mitigation targets, the applicability of the tool for this purpose is limited considering that the multiple underlying assumptions and harmonisation steps involved in the tool and the underlying databases (e.g. population projections, data filling, etc.) make it almost impossible to track with accuracy individual targets and analyse their evolution over time (e.g. changes in assumptions in the base year emissions, qualitative improvement in transparency or other elements, etc.).

As it might be difficult for non-researchers to run the model, we limited the audience more specifically (Sect. 2 + 4), but indicated the use of provided results to stakeholders. The tool is intended to give results on different levels: on country-level (one csv-file providing the single target quantifications, and one providing the derived mitigation pathways), and on aggregated level (pathways for groups of

countries). By keeping the same tool settings and changing only one aspect (e.g., a country's emissions baseline or its reduction ambition), we can track changes on country-level. We indicate this more clearly in Section 3.5 now.

Regarding the style, I agree with the other reviewer regarding the fact that the manuscript is written in a way that is more suitable for model documentation or user manual than a scientific journal, and would therefore suggest the authors move large parts of the main body to an annex, or supplementary information and instead expand the results and discussions sections and focus them on practical applications of the tool (e.g. comparison of the first and second round of NDCs, evaluation criteria or ranking for NDCs). However, considering the focus and target audience of the journal, which relate to modelling (outside my personal expertise) I consider the manuscript can be published in this journal subject to minor revisions, along the lines of my comments above.

As indicated above, we shortened the main body, expanded Section 3.5 on other possible use cases, and extended the discussions in Section 4. Furthermore, in Section 3.4 we additionally include global estimates based on the updated NDCs submitted by 31st December 2020 now to showcase how the aggregated ambition changed over time.

For **Figure 1**: **CRF: 42 countries** (AUS, AUT, BEL, BGR, BLR, CAN, CHE, CYP, CZE, DEU, DNK, ESP, EST, FIN, FRA, GBR, GRC, HRV, HUN, IRL, ISL, ITA, JPN, KAZ, LIE, LTU, LUX, LVA, MLT, NLD, NOR, NZL, POL, PRT, ROU, RUS, SVK, SVN, SWE, TUR, UKR, USA); **BUR: data compared for 0 countries**; **UNFCCC: 25 countries** (ARE, ARG, ARM, BGD, BIH, BOL, BRA, COL, CRI, DOM, GEO, GHA, GUY, IDN, ISR, KGZ, KOR, MDA, MKD, MNG, MUS, PRK, TJK, URY, UZB); **FAO: 188 countries** (AGO, AIA, ALB, AND, ANT, ARE, ARG, ARM, ATG, AUS, AUT, AZE, BDI, BEL, BEN, BFA, BGD, BGR, BHR, BHS, BIH, BLR, BLZ, BOL, BRA, BRB, BRN, BTN, BWA, CAF, CAN, CHE, CHL, CHN, CIV, CMR, COD, COG, COK, COL, COM, CPV, CRI, CUB, CYP, CZE, DEU, DMA, DNK, DOM, DZA, ECU, EGY, ERI, ESP, EST, ETH, FIN, FJI, FRA, FSM, GAB, GBR, GEO, GHA, GIN, GMB, GNB, GNQ, GRC, GRD, GTM, GUY, HKG, HND, HRV, HTI, HUN, IDN, IND, IRL, IRN, IRQ, ISL, ISR, ITA, JAM, JPN, KAZ, KEN, KGZ, KHM, KIR, KNA, KOR, KWT, LAO, LBN, LBR, LCA, LIE, LKA, LSO, LTU, LUX, LVA, MAR, MDA, MDG, MDV, MEX, MKD, MLI, MMR, MNE, MNG, MOZ, MRT, MUS, MWI, MYS, NAM, NER, NGA, NIC, NIU, NLD, NOR, NPL, NZL, OMN, PAK, PAN, PCN, PER, PHL, PLW, PNG, POL, PRK, PRT, PRY, ROU, RUS, RWA, SDN, SEN, SGP, SLB, SLE, SLV, SOM, SRB, STP, SUR, SVK, SVN, SWE, SWZ, SYR, TCA, TCD, TGO, THA, TJK, TKM, TLS, TON, TTO, TUN, TUR, TUV, TWN, TZA, UGA, UKR, URY, USA, UZB, VCT, VEN, VGB, VNM, VUT, WSM, ZAF, ZMB, ZWE).